# Testing a failure surface prediction and deposit reconstruction method for a landslide cluster that occurred during Typhoon Talas (Japan)

Michel Jaboyedoff[1], Masahiro Chigira[2], Noriyuki Arai[2], Marc-Henri Derron[1], Benjamin Rudaz[1], and Ching-Ying Tsou[3]

[1]

[1]University of Lausanne, ISTE-FGSE, ISTE, Lausanne, Switzerland,

[2]Disaster Prevention Research Institute, Kyoto University Gokasho, Uji 611-0011, Japan,

[3]Department of Agricultural and Environmental Engineering, Faculty of Agriculture and Life Science, Hirosaki, Japan.

*Correspondence to*: Michel Jaboyedoff (michel.jaboyedoff@unil.ch)

**Abstract.** The reconstructions of failure surfaces (prior to potential landslides or after their release), landslide deposits or other palaeotopographic features are important for hazard and erosion assessment. The volumes involved in the landslide and failure surfaces constrain the propagation of a landslide, and the knowledge of the past topography helps to understand these hazards. Some methods exist to characterise landslide geometry, but these methods usually require monitoring information. This study tries to assess the validity of the Sloping Local Base Level (SLBL) method for this purpose. Two sets of airborne LiDAR digital elevation models (DEMs) of the Kii peninsula (Japan) are used: the first one was acquired before Typhoon Talas, and the second one was acquired after. A total of 70 deep-seated landslides occurred during this event between 2 and 5 September 2011.

This study shows that the SLBL method is efficient using either the slope deformations identifiable on the DEM before the release of the landslide or a reliable 2.5D failure surface created by using both DEMs (the 2.5D corresponds to a surface which has one and only one z value for each x-y coordinates, in other words, no true vertical topography or overhang can be represented perfectly). In addition, this method allows the reconstruction of eroded deposits and buried valleys. Most of the volumes estimated are within ±35% of the estimation made by Chigira et al. (2013), and the coefficients of expansion range from 10 to 25%. These results show the considerable sensitivity to the parameters used for the reconstruction of the landslide volume estimations and demonstrate the need for an efficient and fast tool to reconstruct potential landslide geometries or histories.

## 1    Introduction

Landslide is an important underestimated threat for society. From 2014 to 2010, 32'322 fatalities occurred worldwide caused by non-seismic landslides (Petley, 2012). Many landslided are the cause of deaths during typhoons (see below) and earthquakes. For instance, the coseismic landslides of the Wenchuan earthquake killed more than 20'000 people in 2008 (Tang

and van Westen 2018). Assessing the landslide volumes is important for both sedimentary budget calculations (Hovius et al., 1997) and for hazard assessment by providing volume frequency distributions (Dussauge-Peisser et al., 2002). The volumes can be estimated based on DEM and assumptions made about the failure surface geometry. We present here a method to estimate landslide volumes.

Only a few authors (Hutchinson, 1983) have proposed so far the 3D reconstruction of failure surface based solely on surface information without underground information.

The half-ellipsoid approximation was proposed for volume estimations (WP/ WLI, 1990; Cruden and Varnes, 1996). This principle has been extended to generate ellipsoidal (Marchesini et al., 2009) or modified ellipsoid (Xie et al., 2004; Nikolaeva et al., 2014) failure surface shapes. Other methods based on observed surface features allow the deduction of a 2D failure

surface shape (Carter and Bentley, 1985; Cruden, 1985), which can be extended into 3D, as shown by the work of Baum et al. (1998). Such approaches, following hypotheses about the rheology of the landslide material, have led to new results in this area of study (Booth et al., 2013; Aryal et al., 2015).

In this paper, we explore the capability of the Sloping Local Base Level (SLBL) method to provide models of the 3D failure surfaces of landslides based on digital elevation models (DEMs) (Jaboyedoff and Derron, 2005; Jaboyedoff et al., 2009). The

landslide cluster that occurred during Typhoon Talas, which hit Japan from 2 to 5 September 2011, was documented by two DEMs, one collected before and one collected after the event. This event induced more than 70 deep-seated landslides on Kii peninsula, unfortunately killing 56 people (Chigira et al., 2013). Most of the landslides were triggered within areas that displayed large precursory slope deformations, which were controlled by sliding and wedge-shaped discontinuities or buckle folding (Chigira et al., 2013), as evidenced by field investigations and the analysis of high-resolution topography data from

these landslides. Aerial LiDAR provided high-resolution DEMs with a 1 m resolution before (pre-DEM) and after (post-DEM) the events (data from the Nara Prefectural Government and the Kinki Regional Development Bureau of the Ministry of Land, Infrastructure, Transportation, and Tourism). This extraordinary opportunity allows us to test methods to reconstruct the (1) failure surface geometries prior to the catastrophic event, (2) buried valley topographies and/or (3) deposit surfaces. We tested the SLBL method on 5 deep-seated landslides that occurred during Typhoon Talas (Akatani, Kitamata, Nagatono, Shimizu

and Akatani-east; see Chigira et al., 2013). The SLBL solution corresponds to a quadratic surface with a constant second derivative along the x-z and y-z planes (Jaboyedoff and Derron, 2005; Jaboyedoff et al., 2009). This shape can be determined based on the knowledge of the length of the landslide and its maximum thickness.

We used mainly hillshade DEMs, slope maps and COLTOP schemes (Jaboyedoff et al., 2009) to define the limits of the landslides and to interpret their structures. Different attempts were performed to reconstruct the failure surfaces and deposits

depending on various subsets of a priori knowledge. Basically, the morphological features extracted from the pre-DEM were used to delineate the limits of the landslides. The main characteristics of the failure surface were obtained by "expert" interpretations and used to calculate the SLBL solutions for different a priori knowledge and scenarios.

In this study, we show the efficiency of the SLBL method as a tool to quickly estimate failure surface geometries without much a priori knowledge. Furthermore, the palaeotopography reconstruction examples indicate that SLBL can be a useful tool to analyse bedrock glacial valleys filled with sediments (Jaboyedoff and Derron, 2005) or the topography prior to a landslide scar.

<a id="5"></a>

## 2 Event and geological setting (after Chigira et al., 2013)

### 2.1 General overview

Typhon Talas occurred in Japan from 2 to 5 August 2011, focused on the Kii peninsula south of Osaka (Figure 1), killing 56 people. The total rainfall between 31 August and 5 September exceeded 1,000 mm, reaching 2,439 in Kami-Kitayama. The average yearly precipitation is 1,300 mm/y in the northwest and more than 3,000 mm/y in the SE of this area (Chigira et al., 2013). The Typhon triggered more than 70 deep-seated landslides. In the past, similar events affected Japan at Kii Mountain in 1889 and at Kyushuin in 2009 (Chigira, 2009).

The region is part of the Cretaceous to Miocene Shimanto accretionary complex. The complex consists of foliated mudstone, sandstone, acid tuff, chert, and greenstones (Kumon et al., 1988; Hashimoto and Kimura, 1999). It is dominated by the so-called "Broken Formation" and mixed rocks with a block-in-matrix fabric (Hashimoto and Kimura, 1999; Festa et al., 2010).

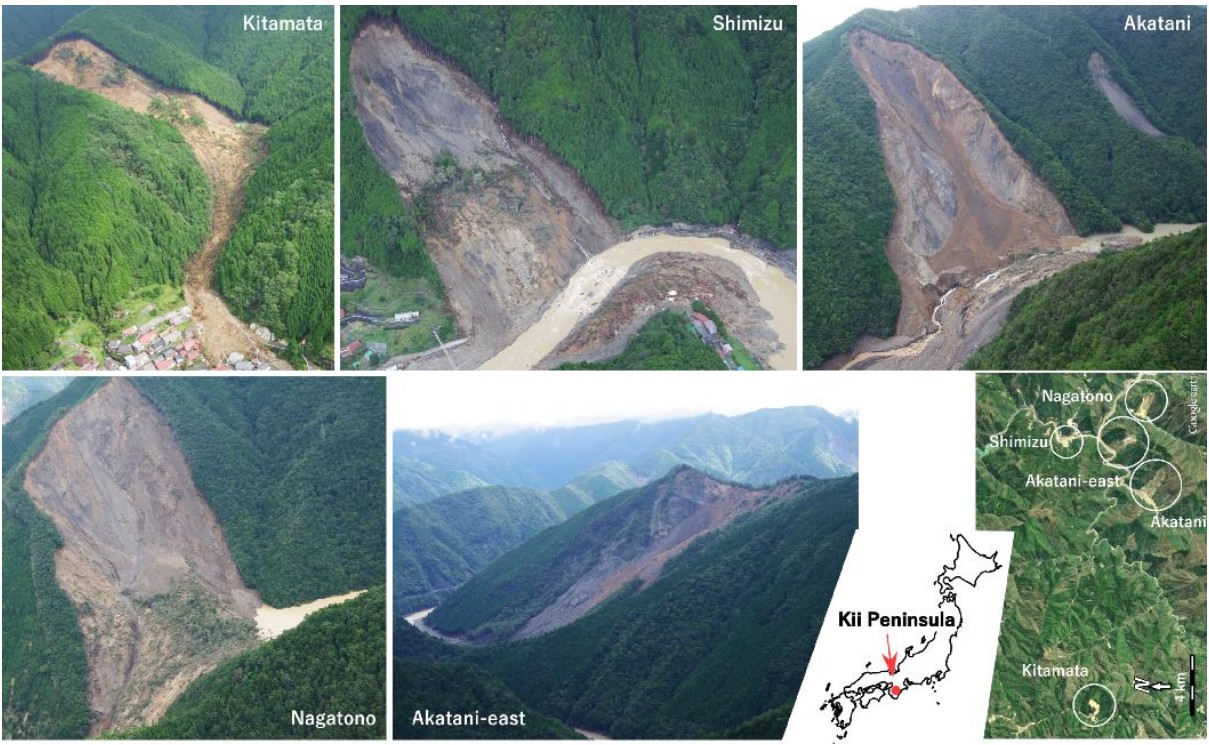

Figure 1: Photographs of the five landslides studied, taken by M. Chigira just after Typhoon Talas. The locations of the photographs are also indicated (images taken by Chigira and partly extracted from Chrigira et al., 2013).

## 2.2 Descriptions of landslides

Chigira et al. (2013) classified the landslides considering the structural origins of the instabilities. All the landslides described here presented important pre-event signs of movement and are predominantly controlled by an unfavourable structural setting. The five landslides we investigated belong to the Hidakagawa Group of the Cretaceous Shimano belt. The rocks are mainly Broken Formation composed of sandstones or mudstone blocks that can include cherts, acid tuffs, greenstones in the mudstone matrix, which is foliated to some degree. The main features of the five landslides described below (Figure 1) are presented in Chigira et al. (2013) and Arai and Chigira (2018).

### 2.2.1 Kitamata

The Kitamata landslide developed on the ridge of a southwest-facing slope. It is roughly laterally limited by two streams. The landslide scar is approximately 280 m long horizontally, 140 m high and 200 m wide. The deposit material is heavily weathered and fractured rock. The main scar is associated with minor undulating faults generally oriented subparallel to the bedding or foliation. The regional foliation dips steeply to the southwest, but the body of the landslide seems to be affected by a flexural toppling, inducing foliation dipping 37° towards the northeast. The flexural toppling defines the failure surface. Before the landslide occurred, a linear depression was observable on the pre-DEM at the top of the ridge parallel to the faults.

### 2.2.2 Shimizu

The Shimizu landslide is located on an east-facing slope. Its scar is approximately 250 m long horizontally, 170 m high and 200 m wide. The debris from this landslide reached the other side of the Kumano-gawa River; this landslide destroyed houses, and 11 people were killed or swept away. The river was dammed for 1 hour and 20 minutes. The landslide material was composed of fractured rock, including sandstones and mudstones. The foliation is generally oriented towards the north, dipping 20° to 35°, and the bedding dips 15° to 20° towards the northeast at the northern margin. Some minor faults trending east-west and dipping 60° towards the north bound the southern margin. The faults, together with bedding and foliation, formed wedges that controlled the direction of sliding. The pre-DEM permitted the identification of 50-m-wide scarp at the top extending along the south side of the future failure crown, which is assumed to have a gravitational origin.

### 2.2.3 Akatani

The Akatani landslide is located on a northwest-facing slope dipping at 34° and developed in the Broken Formation made of sandstones, mudstones and a mixed lithology of sandstone blocks within a mudstone matrix. Its scar is approximately 1000 m long horizontally, 600 m high and 300 m wide. Faults and thrusts are the main structures controlling the landslide limits. Recent observations by Arai and Chigira (2018) indicate that the basal surface was developed along a thrust dipping 35° to the northwest, most of which had been hidden by debris just after the landslide. The thrust is cut on both sides by high-angle faults with northwest–southeast strikes. The Akatani rockslide has an inverse-trapezoidal shape in a transversal profile. The southwestern high-angle fault dips 57° to the northeast. The northeastern high-angle fault dips 60° to the southwest. Evidence of the slope deformation, expressed by the two scarps, is visible in the pre-event DEM. Some buckle folds are also observed

and are assumed to have been created by slope deformation. In addition, a pre-landslide failure occurred at the toe of the landslide.

### 2.2.4    Akatani-east

Sliding surfaces with slickensides were exposed in the landslide scar of the Akatani-east rockslide. These sliding surfaces were identified along the same thrust fault that hosts the Akatani rockslide and was subparallel to the slope before the landslide occurred. Its scar is approximately 700 m long horizontally, 400 m high and 300 m wide. Geological profiles of the landslide suggest that the Kawarabi thrust was exposed along the Kawarabi River before the landslide. Arai and Chigira (2018) found a northwest-southeast trending, northeast-dipping high-angle fault with a 70-cm-wide crush zone after the 2011 landslide. This fault was identified before 2011 at the foot of a steep slope near the top of Mt. Hinose and thus bordered the eastern side of the 1889 landslide. To the east of this fault, along the eastern border of the 2011 landslide, there are two parallel joint surfaces with a strike of N41°W and dip of 41°SW. These joints bordered the eastern side of the 2011 Akatani-east rockslide. These joints and thrust fault created a sliding wedge.

### 2.2.5    Nagatono

The Nagatono landslide is located on a northwest-facing slope with an angle of 34°. Its scar is approximately 560 m long horizontally, 400 m high and 300 m wide. The bedrock exposed in the landslide scar is mudstone-dominated mixed rock that consists of sandstone blocks in a mudstone matrix, but the slide materials appear to contain a greater proportion of sandstone blocks. The limit of the northeast side is controlled by a northwest-southeast-trending fault dipping towards the southwest, and the southwest limit is controlled by several minor faults. These two groups of structures create wedges that control the failure surface. The effect of the slope deformation before the failure occurred, it is visible in the pre-DEM within the upper part of the landslide, which exhibits several scarps dipping between 38° and 45°.

## 3    Methods

### 3.1    COLTOP scheme

The use of high-resolution DEMs allows the analysis of the geological structures that shape the topography. One approach is to use the COLTOP scheme, which provides colours for each planar orientation of the topography or 3D surface using the location of the pole in a stereonet (Jaboyedoff et al., 2009). This permits the identification of visually similar orientations from a DEM. This representation replaces the classical representation that requires two maps, slope angle and aspect, with a single map. This colour scheme is based on the hue-saturation-intensity (HSI) colour wheel integrated in a Schmidt-Lambert stereonet (Jaboyedoff et al., 2009).

### 3.2    SLBL method
#### 3.2.1    Principle

The principle of the SLBL method is an evolution of the base level defined in geomorphology (Mills, 2003) and applied to landslides. This approach allows calculation of the surface above which a rock mass is assumed to be erodible. The SLBL

principle is similar to the principle of the isobase, originally defined by Filosofov (1960) and applied by Golts and Rosenthal (1993). SLBL is rather simple in principle and originates from the signal treatment used in X-ray diffraction (XRD) methods to determine the background signal (Sonneveld and Visser, 1975). For a peak in a curve, the goal is to find the background value, i.e., a signal that continues from the right to the left of the peak. This background identification is performed using an

5 iterative process. Starting from the original values of an array equally spaced by $\Delta x$ in either 1D or 2D, the value of $z_{ij}$ (or $z(x)$ in 1D) is replaced by a lesser value if the average of its neighbours is less than $z_{ij}$. When applied to a gridded DEM, this principle can obtain solutions for the failure surface of landslides (Figure 2). First, the perimeter of the instability must be defined. Then, an iterative process excavates "numerically" a grid DEM ($z(t)_{ij}$). The surface is computed using the following algorithm (Jaboyedoff et al., 2009):

1. For each grid node and at each iteration $t$, an "average" of all the altitudes ($f(z_n(t-1)_{ij}$ is estimated; $z_n$ denotes a set of neighbours, four for a grid (n=4), of the previous iteration $z(t-1)_{ij}$ of the points minus a positive constant $C$ (tolerance):

$$z_{temp}(t)_{ij} = \left(f\left(z_n(t-1)_{ij}\right) - C\right) [1]$$

2. This calculation is performed either by a simple averaging of a given number of neighbours or by fitting a surface.
15 This approach creates a new grid that considers that

$$\text{if } z_{temp}(t)_{ij} < z(t-1)_{ij}, \text{ then } z(t)_{ij} = z_{temp}(t)_{ij} \ [2]$$

otherwise, the value is unchanged. An additional condition can be added to prevent the slope angle of the failure surface from dropping below a given threshold. Similarly, implementing a minimum altitude threshold value can ensure that the new surface does not fall below a certain limit defined by another DEM or a local feature.

3. The iteration is repeated until all the differences ($z_{temp}(t-1)_{ij} - z(t)_{ij}$) over the whole grid are less than a given threshold. The end of the calculation can also be determined by the total volume change between two successive surface iterations.

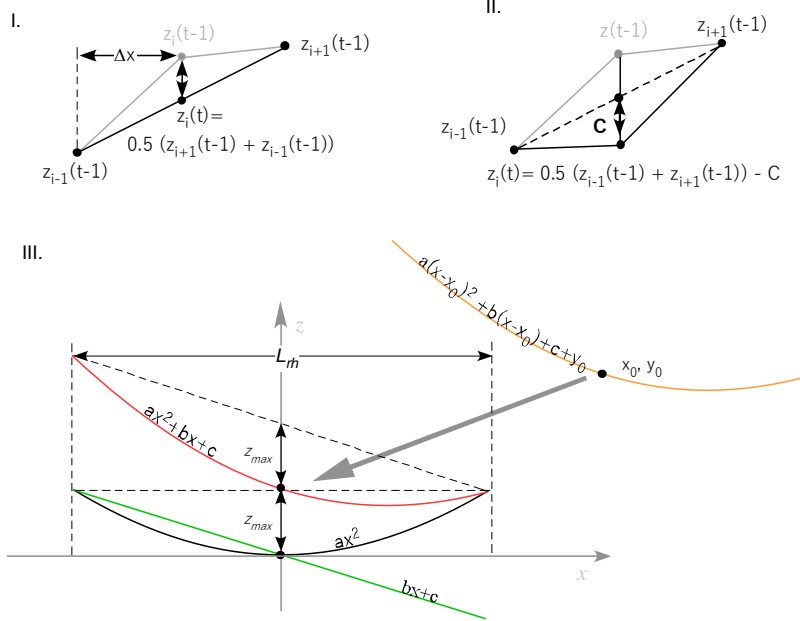

5. **Figure 2: Principle of the calculation of the SLBL. I. The iterative calculation replaces $t$ at each step with the value of the average of its two neighbours if with index $z(t-1) > (z_{i-1}(t-1) + z_{i+1}(t-1))/2$. II. Similar to I., but for a condition of $z(t-1) > (z_{i-1}(t-1) + z_{i+1}(t-1))/2 - C$. III. Diagram explaining why $z_{max}$ is dependent on only $a$ and not on the linear part of the expression ($bx+c$).**

Note that it is also possible to fill holes or concave topographies (landslide scarp, eroded deposits, etc.) by changing the sign of $C$ and reversing the condition in point 2 above. The results of the SLBL calculation depend on $C$, which induces a surface that has some characteristics of a parabola, i.e., its second derivative is constant. Consequently, in 1D, it is possible to link the tolerance $C$ with the parabola coefficient $a$. Assuming a parabolic equation centred on 0 ($z = a\,x^2$), the second derivative is given by $z'' = 2a$. For example, for points with a spacing of $\Delta x$, we have

$$a = \frac{C}{\Delta x^2} \ [3]$$

### 3.2.2 Link between landslide geometry and tolerance $C$

It is important to be able to make the link between the expected shape of a landslide failure surface and the $C$ value. By estimating the horizontal length $L_{rh}$ (projection of the length $L_r$ defined by Cruden and Varnes (1996) on a horizontal plane) or width of the landslide in relation to its maximum vertical thickness $z_{max}$ of the profile, the ratio $e$ can be defined (Figure 2):

$$e = \frac{z_{max}}{L_{rh}} [4]$$

The width corresponds to the width along a defined cross-sections, in order to evaluate the second derivative, but it is close to $W_r$ defined by Cruden and Varnes (1996). Assuming that the cross-sections follow a parabola, we choose to locate the origin of the parabola at the position of $z_{max}$; removing the linear part of the parabolic equation, we obtain

$$z_{max} = a \frac{L_{rh}^2}{2^2} \quad [5]$$

Therefore,

$$a = 4 \frac{z_{max}}{L_{rh}^2} = 4 \frac{z_{max}}{L_{rh}} \frac{1}{L_{rh}} = 4 \frac{e}{L_{rh}} \quad [6]$$

Then, *C* can be written by combining equations 1 and 4:

$$C = a \, \Delta x^2 = 4 \frac{e}{L_{rh}} \Delta x^2 \quad [7]$$

In the case of an inventory of landslides with similar failure surface shapes, the value *C* from equation (7) can be written as

$$C = 4 \, k \, \frac{e}{\sqrt{A}} \Delta x^2 \quad [8]$$

where *A* is the surface area of the landslide (horizontally) and *k* a shape factor, i.e., a constant to be defined depending on the

elongation of landslides. k = 1 implies that the average diameter is used instead of $L_{rh}$. If the horizontal landslide surface is assumed elliptic then $k = \sqrt{\pi w/(4 L_{rh})}$.

### 3.2.3   Example of C value sensitivity

To illustrate the effect of the C value on the volume and $z_{max}$, a simple example is presented: a slope with a slope angle of 35° including an elliptic landslide with $L_{rh}$ = 600 m and width *w*=300 m (Figure 3). In fact the slope has no effect on the volume

since $L_{rh}$ is used. Remember that the second derivative of the parabola profile can be deduced from eq. 7, i.e. $a = C/\Delta x^2$. In the case of an elliptical landslide, *a* value depends on the axis. To compute *a* for the SLBL, the average $a = (a_L + a_w)/2$ is used, where $a_L$ is the parabola coefficient of the long axis and $a_w$ the parabola coefficient of the short axis. As the ratio of axis is equal to 2, $a_w/ a_L = 4$ because of eq. 6 ($a_L$=2×a/5). The results of the SLBL for an elliptic landslide contour the surface obtained is an elliptic paraboloid, as we checked numerically. As a consequence the volume is given by:

$$V_{ep} = \frac{1}{2} \pi \frac{L_{rh}}{2} \frac{w}{2} z_{max} \quad [9]$$

With $z_{max}$ given by eqs. 4 and 5 using the long axis:

$$z_{max} = \frac{2}{5} \frac{a}{4} L_{rh}^2 = \frac{1}{10} \frac{C}{\Delta x^2} L_{rh}^2 \quad [10]$$

Thus the volume is given by:

$$V_{ep} = \frac{\pi}{8} w \frac{L_{rh}^3}{10} \frac{C}{\Delta x^2} = \frac{\pi}{8} 300 \frac{600^3}{10} \frac{C}{5^2} = 1.0179 \times 10^8 \times C \quad [11]$$

This demonstrates the linear relationship of the volume and *C* or *a* values if the surface is planar. It shows that in principle the volume increases linearly with C value, but this may also depend on the geometry of the limit and the topography of the landslide.

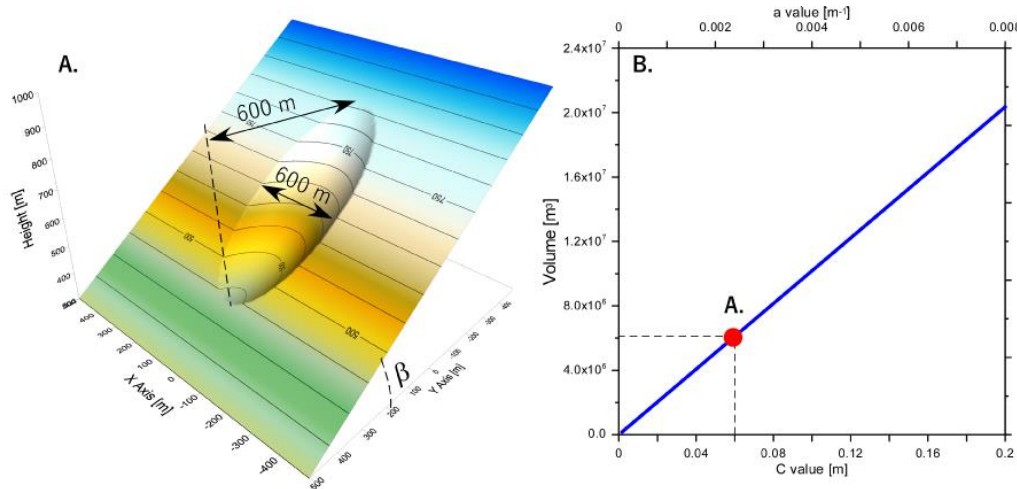

**Figure 3: example of volume values for an elliptic shape and the volume changes with a and C values. A. example of the SLBL for $C=0.06$ m corresponding to $a = 0.00240$ m$^{-1}$ and a $a_L = 00096$ m$^{-1}$ for a volume of $6.1073 \times 10^6$m$^3$. B. Relationship of $a$ and $C$ values with volume for $L_{rh} = 600$ m and the width w=300 m.**

## 4     Reconstruction of failure surfaces and pre-event topographies

The following analysis is based on a LiDAR DEM with a 1 m resolution (1 m DEM). The 1 m DEM was used for geomorphological feature detection. To rapidly compute the SLBL, both the pre- and post-DEMs were resampled at a 5 m resolution (5 m DEM), which is sufficient for reconstructing failure surfaces and deposits because the expected precision does not require more resolution. All the landslide reconstructions were conducted by fully or partially following the process outlined below (Figure 4):

Step 1: Blind analysis: Here, we always start with the pre-DEM, because the first author has tried to define the landslide contours and volumes without knowing the contour given by the post-DEM, based only on hillshade, slope map, etc. obtained from the pre-DEM. This is performed to illustrate the potential and the efficiency of the method to define different scenarios for the failure surface and volume, which may be involved in a future catastrophic failure.

1. The contour of the landslide is defined based on the pre-DEM hillshade, slope map or 3D view. This allows the identification of counter slopes, scarps, and slope breaks that are connected to former landslides scars or cracks. Lineaments defined by changes in slope orientation are also identified.
2. $C$ is defined based on longitudinal or transverse cross-sections made by hand, permitting an estimation of $e$ and $L_{rh}$.
3. Cross-sections of the SLBL are extracted from the model, and $C$ is adjusted to obtain the best visual results.
4. The volume is calculated and compared to the results of Chigira et al. (2013), which were obtained from multiple cross-sections, and the $C$ value can be adjusted if the results are too dissimilar.

Step 2: Analysis based on the pre-and post-1-m DEM (mixed scenario or MS)

1. The contour is re-analysed based on the pre- and post-DEM hillshades, slope map or 3D view and map of the elevation differences.
2. The process is then identical to that in the 1$^{st}$ step from point 2.

Step 3: Eroded deposit reconstruction — if the deposit has been eroded, the SLBL method is used to reconstruct the immediate post-event topography (before any human- or river-derived erosion occurs)

1. The contour of the missing zone is defined based on the difference between the pre- and post-1-m DEMs.
2. The reconstruction is based on trial and error, using the cross-section of the reconstruction. Note that $C$ is usually close to zero.
3. The volume is computed; if inconsistent results are obtained, the workflow returns to point 1. If the deposit is not fully included within the DEM, simple geometrical rules are applied to calculate and add the missing volumes.

Step 4: Calculation of the deposit

1. The reconstruction of the total volume is based on the post-DEM, reconstructed or not, and the SLBL is constructed from the pre-DEM. The deposit contour is defined, providing an expansion factor. In some cases, the volume remaining under water behind the dam created by the landslide or outside the DEM is estimated by simple geometrical rules.

Step 5: Reconstruction of the pre-topography based on the post-DEM and/or mixed scenario

1. This step follows the same scheme as that of step 2, but the inverse SLBL and volume computation are based on the post-DEM. In this case, the verification is valid only where the deposit does not change the altitude of the landslide limits or if the pre-DEM is used.

Step 6: Reconstruction of the topography before a deposit

1. The contour of the deposit is determined based on the post-DEM.
2. The value of $C$ changes depending on the configuration.
3. The volume is computed, and if inconsistent results are obtained, the workflow returns to point 1.

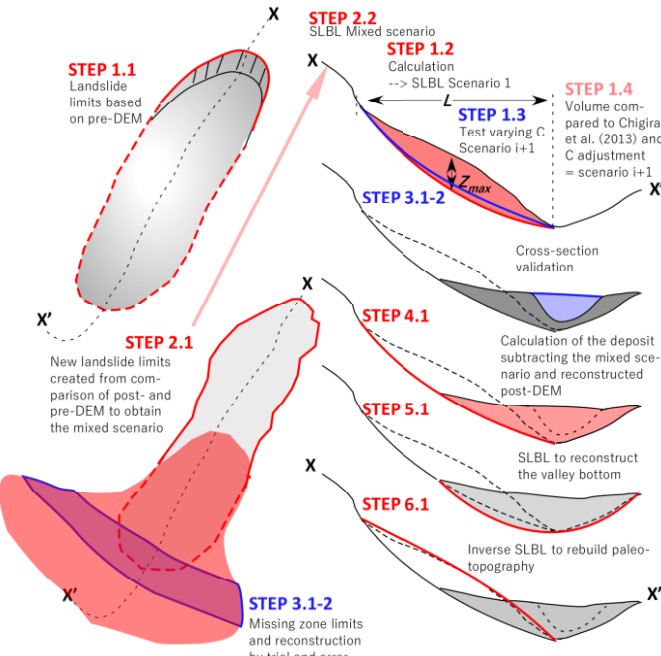

**Figure 4: Illustration of the different steps used to assess the volumes, the failure surface and the palaeotopography using an SLBL.**

### 4.1 Kitamata landslide

#### 4.1.1 Step 1: Three scenarios based on pre-topography

The Kitamata reconstruction shows that, depending on the chosen landslide limits based on the pre-DEM, the volumes obtained for the three proposed scenarios (Step 1) vary (Table 1; Figure 5). The first scenario takes into account only the spur formed by the topography ($0.318 \times 10^6$ m$^3$), with $C = 0.05$ deduced from the *a* parameter (0.002) of the parabola drawn by hand. For the two other scenarios, a trial-and-error procedure was used by taking into account the result of scenario 1 and the expected shape of the failure surface to avoid over-deepening. The detection of the upper limit of the scar on the 1-m-pre-DEM hillshade is challenging, so the middle of the flat area was taken as the external upper limit. In fact, the scar of the landslide developed slightly beyond that limit. The second scenario includes limits that are drawn by taking into account past scar contours ($0.810 \times 10^6$ m$^3$). The third scenario takes into account the expected extreme limits of past potential movements ($0.991 \times 10^6$ m$^3$).

| | Kitamata | | Shimizu | | Akatani | | Akatani-E | | Nagatono | |
|---|---|---|---|---|---|---|---|---|---|---|
| | Volume | C (a, Δx) | Volume | C (a, Δx) | Volume | C (a, Δx) | Volume | C (a, Δx) | Volume | C (a, Δx) |
| | [10$^6$ m$^3$] | [m] [m$^{-1}$] [m] | [10$^6$ m$^3$] | [m] [m$^{-1}$] [m] | [10$^6$ m$^3$] | [m] [m$^{-1}$] [m] | [10$^6$ m$^3$] | [m] [m$^{-1}$] [m] | [10$^6$ m$^3$] | [m] [m$^{-1}$] [m] |
| Volume by Chigira et al. (2013) | 0.88 | | 0.93 | | 8.20 | | 2.10 | | 4.10 | |
| **STEP I** | | | | | | | | | | |
| Scenario 1 | 0.318 | 0.05 (0.002, 5) | 1.33 | 0.035 (0.0014, 5) | 18.00 | 0.025 (0.001, 5) | 1.59 | 0.0075 (0.0003, 5) | 3.94 | 0.019 (0.00076, 5) |
| Scenario 2 | 0.810 | 0.075 (0.003, 5) | | | 8.35 | 0.035 (0.0014, 5) | 2.06 | 0.015 (0.0006, 5) | 4.11 | 0.02 (0.0008, 5) |
| Scenario 3 | 0.991 | 0.04 (0.0016, 5) | | | | | | | | |
| **STEP II: prior to collapse** | | | | | | | | | | |
| Mixed scenario based on pre- and post DEM 1 | 0.884 | 0.002 (0.002, 1) | | | | | | | | |
| | 0.885 | 0.045 (0.0018,5) | 0.975 | 0.066 (0.00264, 5) | 11.03 | 0.035 (0.0014, 5) | 1.75 | 0.0075 (0.0003, 5) | 3.19 | 0.02 (0.0008, 5) |
| Mixed scenario based on pre- and post DEM 2 | | | | | | | 2.15 | 0.015 (0.0006, 5) | 4.09 | 0.03 (0.0012, 5) |
| **STEP III & IV: deposit volumes** | | | | | | | | | | |

| | | | | | C (a, Dx) Exp coeff. | | | | | |
|---|---|---|---|---|---|---|---|---|---|---|
| | Volume | Exp coeff. (+V) | Volume | (+V) | Volume | Exp coeff. (+V) | Volume | Exp coeff. (+V) | Volume | Exp coeff. (+V) |
| Volume of deposit | [10$^6$ m$^3$] | [-] [10$^6$ m$^3$] | [10$^6$ m$^3$] | [-] [10$^6$ m$^3$] | [10$^6$ m$^3$] | [-] [10$^6$ m$^3$] | [10$^6$ m$^3$] | [-] [10$^6$ m$^3$] | [10$^6$ m$^3$] | [-] [10$^6$ m$^3$] |
| Simple subtraction (post-DEM - mixed sc.) | 1.09 | 24% | 0.602 | | 11.73 | 6% (-) | 2.42 | 38%(-) | 4.77 | 19% (-) |
| Simple subtraction (post-DEM - sc. X) | | | | | 9.60 | 14.9% | 2.64 | 22%(-) | | |
| Correction for missing deposit (calculation or | | | | 13% (+0.24)- | 12.62 | 12.6% (+0.89) | | | | |
| SLBL) | | | | 15%(+0.34) | | | | | | |
| (post-DEM - sc. 2) | | | | 0-0.035 (0-X,5) | 10.49 | 20.5% (+0.89) | | | | |
| Correction for outcrop in the scar | | | | | 12.62 | 13.9% (MS-0.16) | 2.42 | 3.2% (SC1+.406) | | |
| | | | | | | | 2.64 | 2.9% (Sc2+.599) | | |
| **Step V** | | | | | | | | | | |
| Reconstruction | | 0.944 0.0375 (0.0015, 5) | | 1.03 0.033 (0.00132, 5) | | 8.19 0.035 (0.0014, 5) - | | . | | . |
| **STEP VI and remarks** | | | | | | | | | | |
| Volume within the defined polygon based on the difference between the pre- and post-DEMs | 0.461 | | | | With sc2, the volume of the deposit is 9.60, and adding the missing part of the deposit it give a volume of 10.49, which leads to expansion coefficients of 12.95 and 20.5% | | | | | |
| Volume within the defined polygon based on the difference between the post-DEM and valley bottom recontructed with SLBL | 0.345 0.007 (0.007, 1) | | | | | | | | | |

Table 1: Summary of volumes and SLBL results (see text; SC = Scenario). In the columns the table provides the volumes obtained based on C-values with the corresponding a-value and the grid size used in parentheses. The steps of the method and the way used to compute the failure surface are found along the rows.

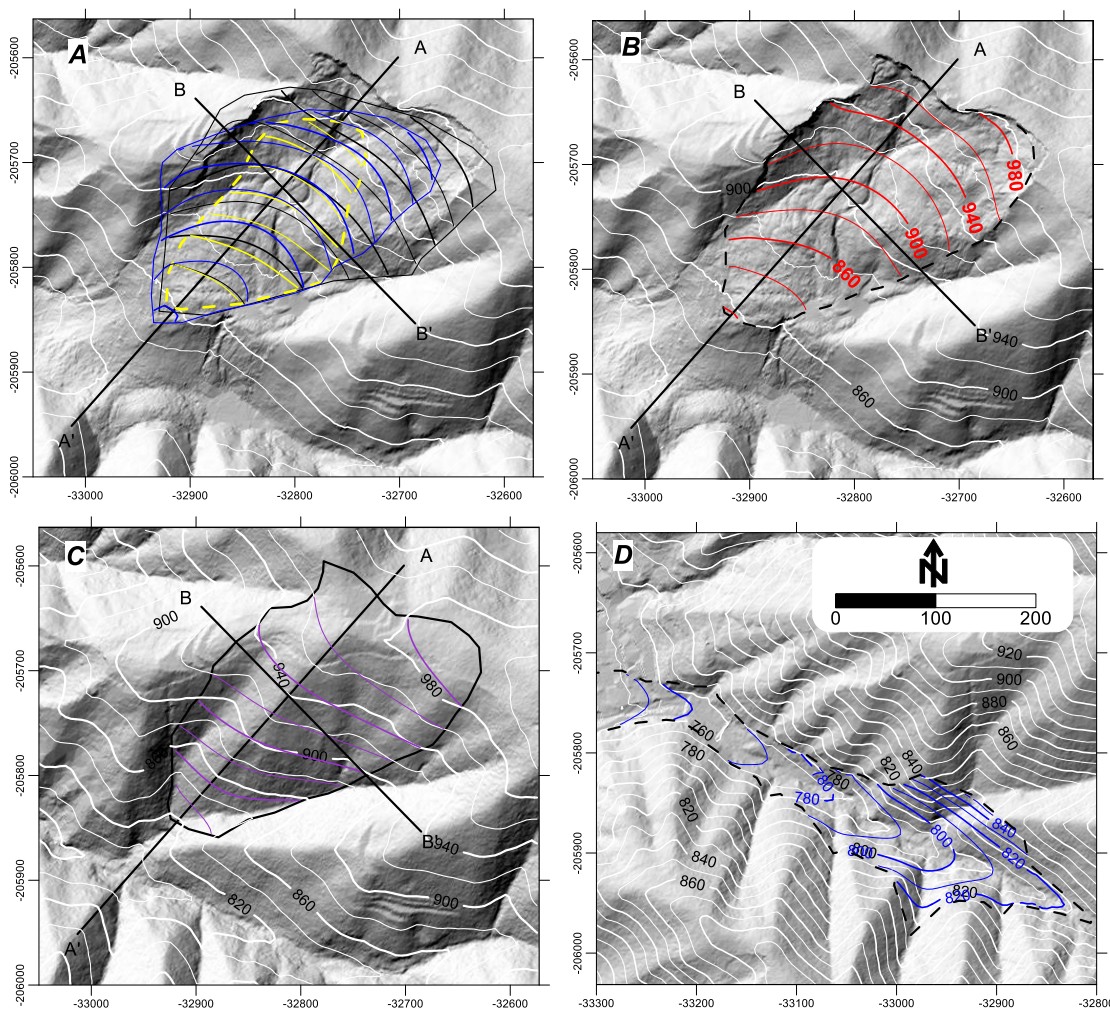

**Figure 5: SLBL applications for the Kitamata landslide: A. Failure surfaces of three scenarios based on the analysis of the topography before the 2011 landslide without post-failure information (SC1: yellow; SC2: blue; SC3: black). B. Mixed scenario in red with the contour deduced from both the pre- and post-DEMs. C. Reconstruction of the topography based on the post-DEM compared to that of the pre-DEM. D. Reconstruction of the valley based on the post-DEM.**

### 4.1.2 Step 2: Landslide limits based on the pre- and post-DEMs and deposit volume

More reliable landslide limits were obtained when both DEMs were considered (mixed scenario) than when just one DEM is used. In the upper part of the slope, the scar from the failure is clear, and in the lower part of the slope below the deposit, the pre-DEM was used to delineate the scar. The result found for $C = 0.002$, equivalent to $a$ in scenario 1 but determined by using the 1 m pre-DEM with the limits deduced from the difference between the pre- and post-DEM, exhibits exactly the same volume as that found by Chigira et al. (2013) ($0.88 \times 10^6$ m³). Using the 5 m pre-DEM, $C = 0.05$; however, we used 0.045 to obtain exactly the same volume. The failure surface obtained fits the topography after the landslide quite well, in the areas where the deposit is not very thick (Figures 5.b and 6). On the northwestern part of the slope, the SLBL is slightly above the

observed surface after the event; therefore, an important planar structure may have also played a role. On average, the confidence of these results is good, and the surface follows the main trends in the upper part of the slope, considering that some shallow deposits can hide the true failure surface, complicating the comparison.

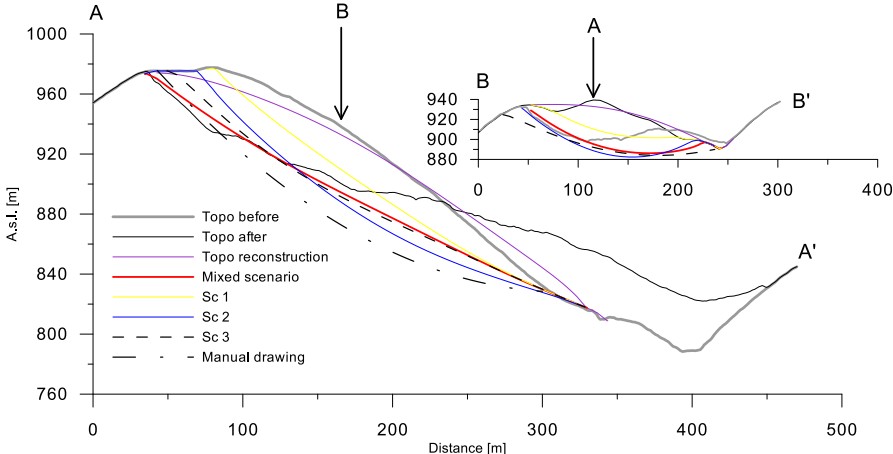

**Figure 6: Cross-section of the different scenarios for the Kitamata landslide failure surface and topography reconstructions using the pre-DEM to constrain the altitudes in order to test the method (Sc # = Scenario X).**

### 4.1.3 Step 4: Deposit volume calculation

There was no need of step 3. Starting from the mixed scenario, it is possible to evaluate the "true" volume of the deposits by subtracting the pre-DEM (including the SLBL) from the DEM acquired just after the landslide. The map of the thickness of the deposits presents a reliable shape with greater thickness at the bottom of the slope (Figure 7). Nevertheless, this result shows that approximately 50,000 $m^3$ of material is missing in the upper northwest part of the slope (due to the SLBL surface). The total volume of the displaced mass, corrected for the negative volumes, reaches 1.09 $Mm^3$, which corresponds to an expansion coefficient of 24%.

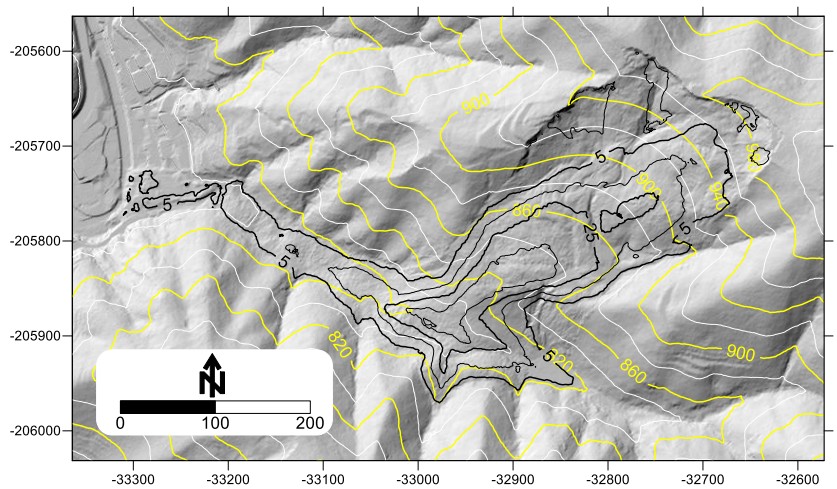

**Figure 7: Thickness of the deposit of the Kitamata landslide based on the comparison of the post-DEM including the mixed scenario.**

#### 4.1.4 Step 5: Reconstruction of the topography anterior to the landslide based on the post-DEM

Using the limits defined by the mixed scenario, the inverse SLBL was used to check the validity of the reconstruction of the pre-event topography; the *a* value of a parabola fitted to the cross-section through the pre-DEM is estimated to be 0.0015, which provides $C$ = 0.0375 for the 5 m DEM. This is not a full test because we used the pre-DEM topography as constraints in order to avoid an intermediate step of deposit removal. The results of this test are in good agreement with the present topography, but the results do not match the present topography near the creeks. It is clear that fluvial erosion has affected the

topography (Figure 5.c). The volume obtained by comparing the reconstructed topography with the post-DEM provides a volume of $0.57 \times 10^6$ m$^3$, which is reasonable considering that some deposits remain on the failure surface. This comparison also provides a negative volume of approximately $0.012 \times 10^6$ m$^3$, which likely corresponds to the local mass movements linked to fluvial erosion prior to the landslide. The volume defined by two reconstructions, i.e., the failure surface (mixed scenario) and the reconstructed topography, is $0.94 \times 10^6$ m$^3$, which is close to the assumed real value. Nevertheless, both

positive and negative volumes are calculated because the results are not perfect, but these errors are balanced, making the results relevant.

### 4.1.5 Step 6: Reconstruction of the valley before the landslide

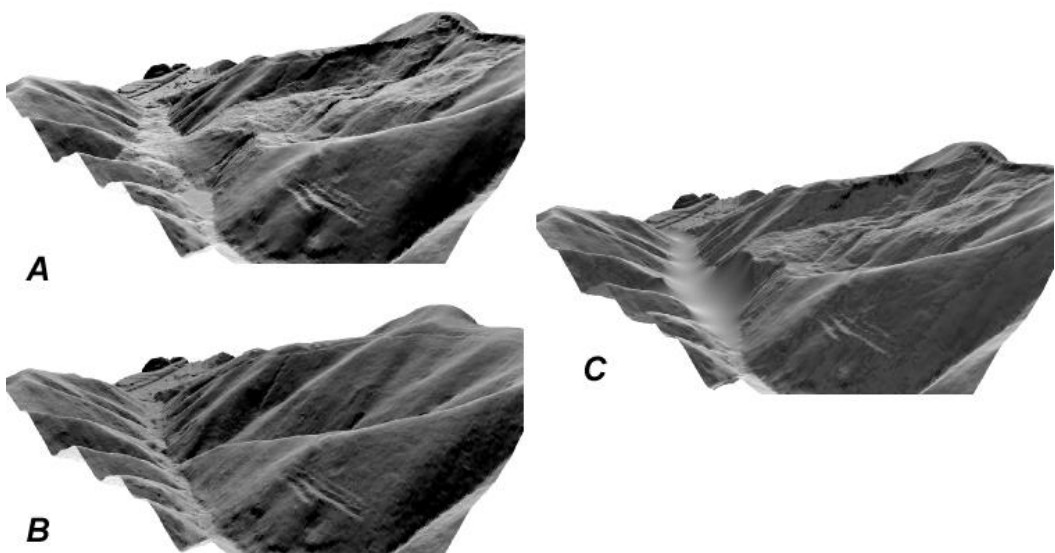

**Figure 8: ArcScene illustration of the difference between the post-DEM (A), pre--DEM (B) and the SLBL reconstruction (C) of the valley filled by the Kitamata landslide.**

To test the validity of creating a valley below the deposit with SLBL, a model was evaluated by delineating the valley with a polygon and applying an SLBL with $a$ = 0.0070, a value obtained by trial and error in order to obtain a linear shape of the longitudinal profile of the valley. The model results are similar to the topography within the flanks of the valley; however, near the river, a considerable difference arises (Figures 5.d and 8). Computing the volume using the SLBL within the polygon gives $0.345 \times 10^6$ m$^3$, while in the same polygon, the difference between the pre-DEM and post-DEM gives $0.461 \times 10^6$ m$^3$, which corresponds to a 20% difference (Table 1: Figure 5) because the SLBL does not result in a sufficiently deep river bottom. Note that slightly more than half of the volume ($0.57 \times 10^6$ m$^3$) seems to remain in the scar, and over $0.461 \times 10^6$ m$^3$ is between the scar and the valley ($0.03 \times 10^6$ m$^3$), for a total of $1.09 \times 10^6$ m$^3$.

## 4.2 Shimizu landslide

### 4.2.1 Step 1: Estimation based on the pre-DEM

The delineation of the limits of the landslide using the hillshade and COLTOP scheme before the landslide is obvious in the upper part of the slope (Figure 9). A manually drawn cross-section of the potential failure surface is used to define the parameter $C$ (Figure 10). The schematic cross-section provides $C = 0.031$ ($L = 400$ m and $z_{max} = 50$ m), but in order to obtain a more realistic curve, $C$ was increased to 0.035 (scenario 1), which led to a volume of 1.33 Mm$^3$ based on the 5 m DEM (Table 1; Figure 11). In the upper part of the slope, the resultant surface follows the contour lines of the post-DEM quite well; however, the volume is too large compared to the 0.93 Mm$^3$ suggested by Chigira et al. (2013).

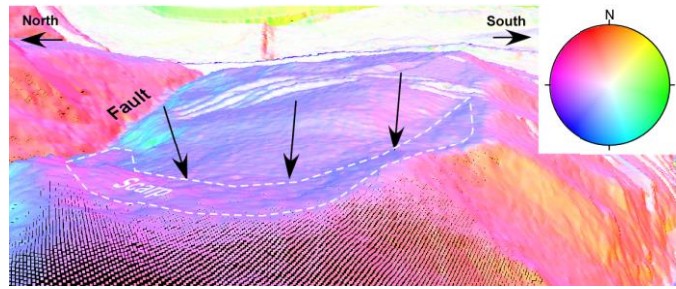

**Figure 9: Scarp of the Shimizu landslide viewed from the top using the COLTOP scheme which provides the colour of the normal to the topography in lower hemisphere. The scarp (area defined in white dashed line) illustrates the precursory movements. The arrows indicate the top of the moved mass.**

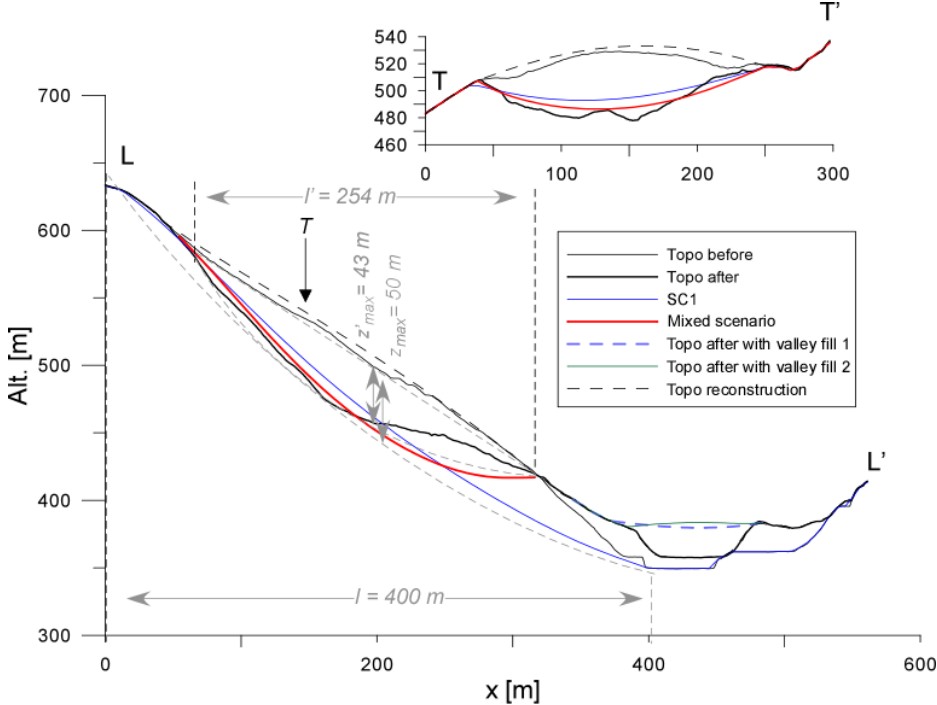

**Figure 10: Schematic cross-section of Shimizu landslide for different scenarios and the manual cross-section determined in order to obtain *e* values (grey lines and text). The two solutions for valley filling are indicated.**

### 4.2.2   Step 2: Reconstruction of the failure surface using both the pre- and post-DEM

The post-DEM clearly shows that the landslide contour before the event includes a too large area in the lower part of the slope.

10   Using the perimeter defined by using both the pre- and post-DEM, the results appear to be very similar to the observations in

the scar, and the cross-section fits the observations well (Figure 11). Using $C = 0.0666$, the volume by this method obtained

($0.975$ Mm³) is very close to the $0.93$ Mm³ obtained by Chigira et al. (2013).

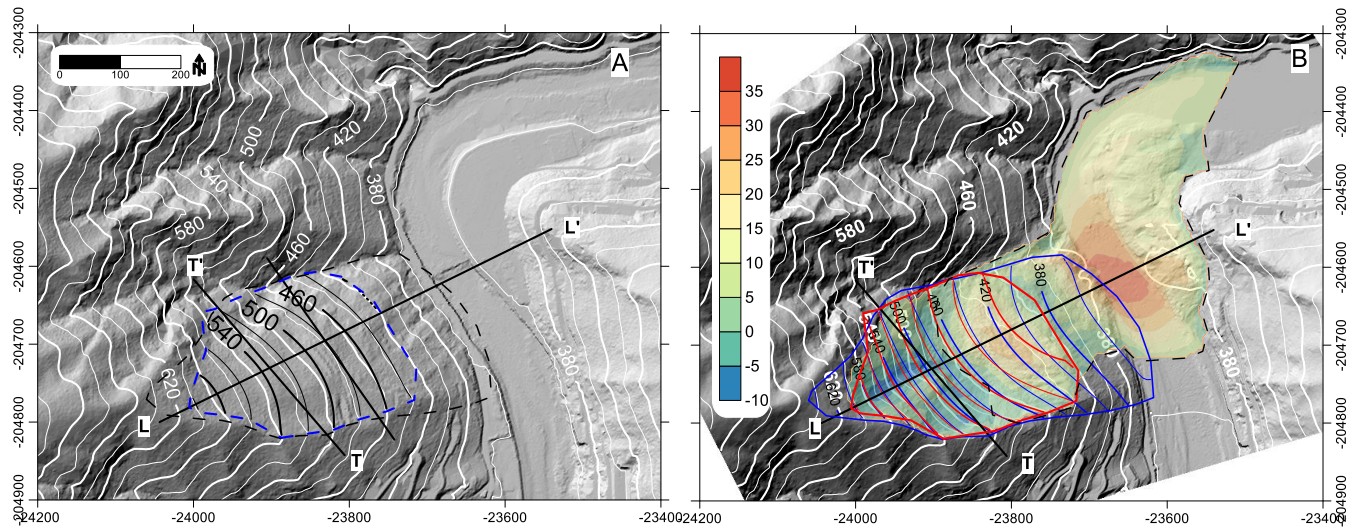

**Figure 11: A reconstruction of the topography of the Shimizu landslide based on the 5 m post-DEM (in black) compared to that DEM. B. Comparison of the SLBL scenarios with the post-DEM and the results of the deposit thickness (colour scale) determined using the difference between the mixed scenario (in red) and 5 m post-DEM; scenario 1 is shown in blue. The dashed line indicates the contour used to define the volume.**

### 4.2.3    Steps 3 and 4: Deposit calculation and its reconstruction

The volume budget of the scar and the deposit using the difference between the pre- and the post-DEM provides a negative volume of 28% (602,100 – 836,400 = -234,300 m³). This result is expected because part of the deposit was washed away by the river; however, this method cannot be applied in a straightforward manner, i.e., the difference between the pre- and post-DEMs (without treatment) shows that the excess volume is smaller than the negative volume. To solve this problem, the inverse SLBL is used to reconstruct the "original deposit surface". The first step in this process is to create a mask that contours the identifiable features of the deposit (Figure 11.b). The deposit topography is reconstructed by two models, based on $C = 0.0$ and 0.035, in order to obtain a reliable surface (Figures 10, 11 and 12). The missing volumes from these two models were 241,000 m³ and 343,000 m³, respectively. Using this reconstructed DEM and subtracting the SLBL (using the contour based on the mixed scenario), the total volume of the deposit is estimated at 1.16 and 1.185 Mm³ for the two models, respectively, representing expansion coefficients of approximately 13% to 15% (Table 1; Figures 11 and 12). Several attempts were necessary to obtain these results due to problems related to the river: first, the 1 m DEMs from before and after the landslide have a 5 m difference in the riverbed. This is likely due to the formation of a deposit caused by river damming and the water level difference. In addition, the high water level of the river during the LiDAR acquisition after the landslide makes finding the limits of the deposit more challenging.

If the low values of expansion are correct, they indicate that the landslide had already moved and the material was already heavily disturbed, broken and expanded when the pre-DEM was recorded. This seems to be confirmed by the morphology visible on the pre-DEM. Clearly, the landslide existed before, and this catastrophic event was a reactivation. In addition, a

major unknown is the role of the river sediments that have moved during landslide propagation; however, it can be expected that their expansion is not significant.

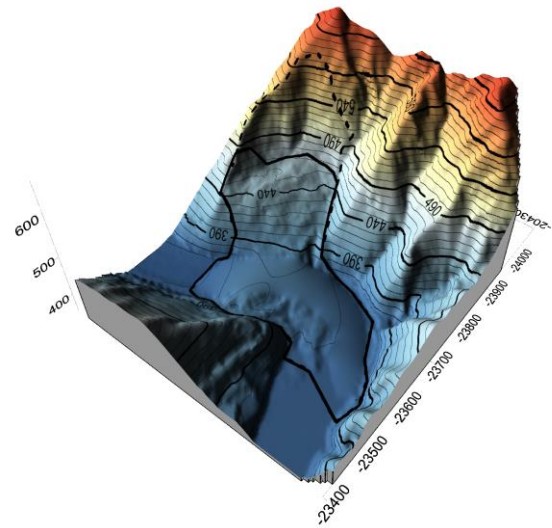

**Figure 12: View of the final result of the deposit reconstruction; the black line indicates the contour used to compute the volume, and the red line indicates the upper scar.**

### 4.2.4    Step 5: Reconstruction of the topography anterior to the landslide based on the post-DEM

To reconstruct the pre-event topography, an attempt was made using half of the $C$ value used for the mixed scenario, i.e., 0.033, similar to the $C$ value used for scenario 1 (Figures 10 and 11a). Subtracting the reconstruction from the post-DEM provides a result of 1.03 Mm$^3$. The budget of material remaining in the scar compared to the SLBL is approximately 0.176 Mm$^3$, which leads to a total estimate of 1.21 Mm$^3$. A comparison of the pre-topography to the reconstruction shows that a volume of 0.26 Mm$^3$ is in excess, probably removed in part by the fluvial erosion of the slope by the northern stream and related small mass movements. The final resultant volume of 0.95 Mm$^3$ is in agreement with the previous results.

Step 6 was not performed for this landslide since the riverbed plays a major role.

### 4.3    Akatani landslide

### 4.3.1    Step 1: Two scenarios based on pre-topography

Using the 1 m DEM collected before the Akatani landslide permits the delineation of the instability contour limits for two possible conditions, which will be referred to as scenarios (Figures 13 and 14). Scenario 1 resulted in the largest volume considered and was one of the possible "extreme events", with a computed volume of 18 Mm$^3$. Then, looking more carefully at the pre-DEM, a second and more realistic scenario was developed. This second scenario was assumed to be the most likely model for the failure surface using $C = 0.035$ (with a 5 m DEM). The volume obtained from scenario 2 is $8.35 \times 10^6$ m$^3$, which approximately corresponds to the volume defined by Chigira et al. (2013) of $8.2 \times 10^6$ m$^3$. In Figure 13, the contours of the SLBL surface are compared to the topography after the landslide. These results show a reasonable agreement.

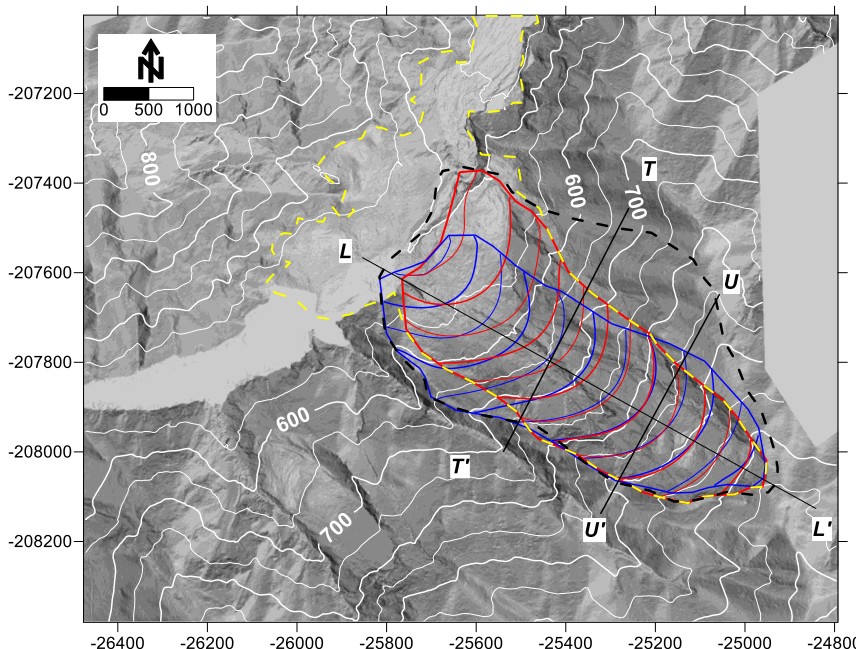

**Figure 13: Aakatani hillshade-like (coltop scheme in B/W) map from the 1 m post-DEM, displaying the limits of the landslide scenarios and failure surface topography: Scenario 1: largest scenario (black dashed line). Scenario 2: the most likely based on the pre-DEM interpretation (blue line); mixed scenario in red. The yellow dashed-line is the limit of scar and deposit. The cross-sections are shown by L-L', T-T', and U-U'.**

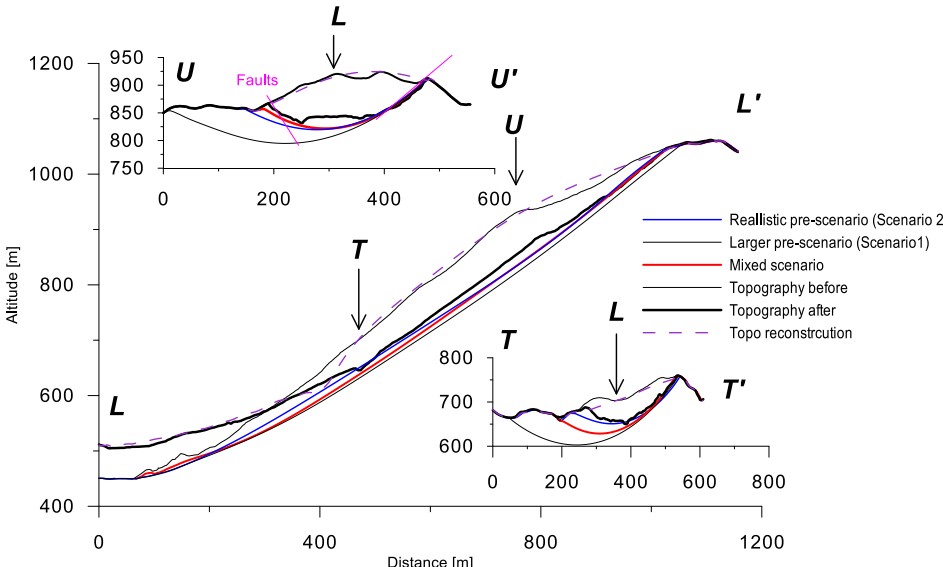

**Figure 14: Cross-sections the Akatani landslide (locations are indicated in the figure 13). The different scenarios indicated in profile U-U' includes a profile of the extreme scenario.**

### 4.3.2    Step 2: Reconstruction of the failure surface using both the pre- and post-DEMs

The mixed scenario was defined based on both the pre-DEM and post-DEM. In the upper part of the slope, the limits are clear, but in the lower part of the slope, the limit is not exact because of the deposit. Applying the same *C* value (0.035), the volume defined reached $11.03 \times 10^6$ m³, which is larger than the results of Chigira et al. (2013) by 34.5%. However, we kept this solution because the post-DEM contours in the upper part of the slope fit the failure surface rather well where the failure surface is nearly outcropping (Figures 13 and 14).

For the volume calculation, a volume was removed from the mixed scenario. From inspecting the slope and hillshade maps in Figures 1, 13, and 15, it can clearly be seen that a rock spur remains after the landslide event in the northeast area of the scar. Its longitudinal section is 3945 m², and its width is 82 m. Assuming a triangular shape, we obtain 3,945 m² × 82 m / 2 = 0.162 × $10^6$ m³.

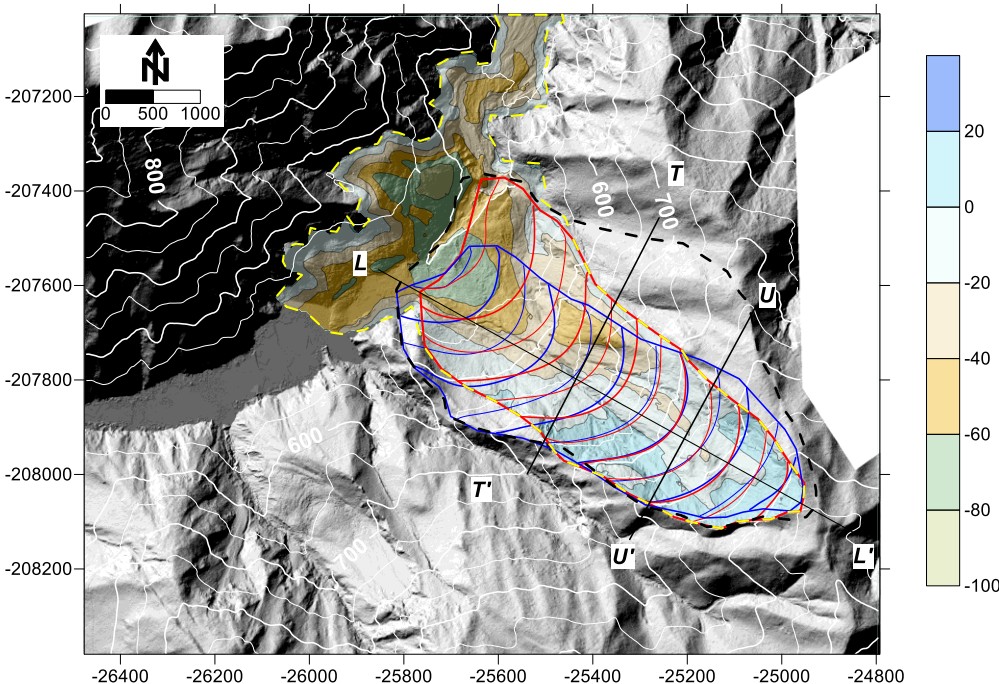

**Figure 15: Aakatani hillshade post-DEM (1 m) and contour (5 m) of the depth of the deposit based on the comparison between the mixed scenario and the post-DEM results.**

### 4.3.3    Step 3: Adding missing volumes

The northern part of the deposit lies within the riverbeds, outside of the DEM. By using the difference between the pre- and post-DEMs the section of the deposit downstream can be estimated (Figure 15). The thickness of the deposit decreases linearly within the map. The linear extrapolation of the longitudinal profile of the deposit topography out of the map intersects the river profile extrapolated at 390 m out of the map. By applying a simple rule using an average section of 2760 m² at 453 m from the assumed zero, the volume is (2760 × 453 / 2) = 0.62 × $10^6$ m³. Upstream of the landslide dam, below the water, using the same

rule as above but with a manual estimate, the volume reaches $166 \text{ m} \times 3200 \text{ m}^2 / 2 = 0.26 \times 10^6 \text{ m}^3$. The total missing from the deposit itself is then $0.89 \times 10^6 \text{ m}^3$.

### 4.3.4    Step 4: Deposit volume calculations

The subtraction of the mixed scenario from the post-DEM provides a volume of $11.73 \times 10^6 \text{ m}^3$, which corresponds to an expansion coefficient of 6% (Figure 15; Table 1). Correcting for the missing deposit $((11.73 + 0.89) \times 10^6 \text{ m}^3)$ gives $12.62 \times 10^6 \text{ m}^3$ with an expansion coefficient of 12.6%, and if the volume of the spur is removed, the expansion coefficient rises to 13.9%. Notably, by using scenario 2 (post-DEM – SC2 = $9.6 \times 10^6 \text{ m}^3$), adding the missing volume results in $(9.6 + 0.89)10.49 \times 10^6 \text{ m}^3$ compared to the $8.35 \times 10^6 \text{ m}^3$ in place (not including the spur), leading to an expansion coefficient of 20.5%.

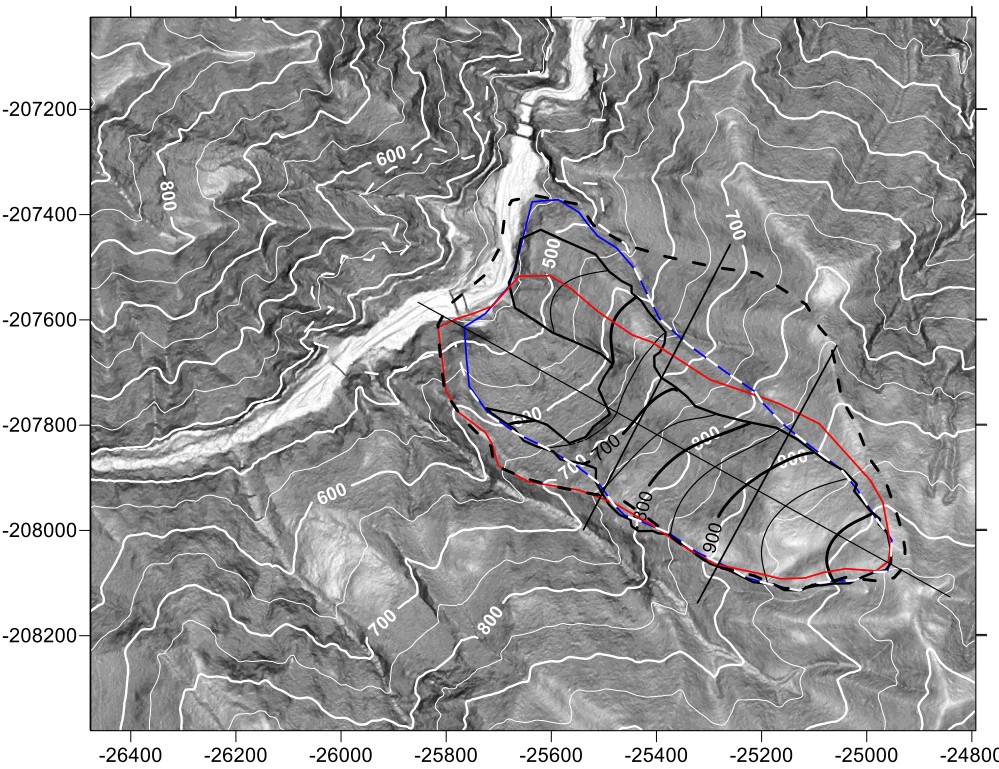

**Figure 16: Slope angle map of the Aakatani landslide area from the 1 m pre-DEM, displaying the reconstruction of the pre-topography (in black) compared to the pre-DEM (in white).**

### 4.3.5    Step 5: Reconstruction of the topography before the event based on the post-event topography

To fill the post-DEM to determine the pre-DEM, the inverse SLBL may be used with a $C = 0.035$, which is similar to the one chosen for the failure surface; the topography before the landslide is reconstructed without information from the DEM acquired before the landslide. However, the cross-section of the topography in the western part of the DEM can allow us to confirm that 0.035 is a reliable value for the calculation of $e$ on the ridge, i.e., creating a topography profile of the spur and calculating $e$.

The limit used to reconstruct the topography takes into account that only the zone with rock outcrops will be rebuilt, avoiding the toe already affected by an ancient scar (see Chigira et al., 2013). The limit is visible in Figure 16. The results are in agreement with the pre-DEM, except that the inverse SLBL does not include erosion by rivers. The volume computed between the post-DEM and the reconstruction provides a surprisingly close value of $8.19 \times 10^6$ m³. However, this is an estimate because in the lower part of the scar, the deposit is present.

### 4.3.6    Structural analysis

The analysis of the 1 m DEM using the COLTOP scheme reveals 3 main structures, which were already identified by Chigira et al. (2013). As shown on Figure 16, the topography before the landslide was already shaped by red (F1) and blue (F2) orientations, which may correspond to regional and local fault sets. The purple shading corresponds to the expression of the thrust shaping the lower part of the failure surface (Arai and Chigira, 2018). F1 is also associated with a yellow orientation in some areas; therefore, these structures are possibly mechanically linked (conjugate).

Figures 17 and 18 clearly show the role of the different structures that shape the failure surface. These structures also control some of the limits of the failure surface. In many places, they often create composite surfaces that generally follow the SLBL (Figure 17).

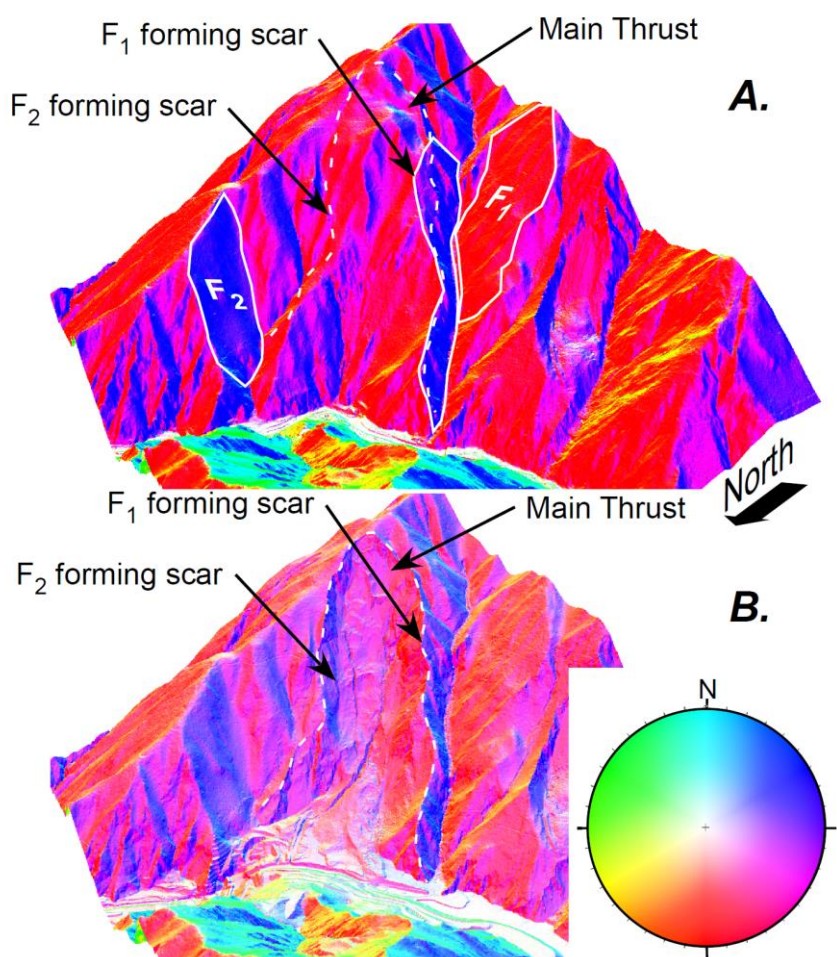

**Figure 17: A. COLTOP colour scheme (the colours of the normal to the topography are given in the lower hemisphere). in ArcScene showing the different structures shaping the topography before the Aakatani landslide. B. Topography after the slide with the same colour scheme, clearly showing the control of the failure surface by these main features. The structures that were difficult to infer in A are well developed here.**

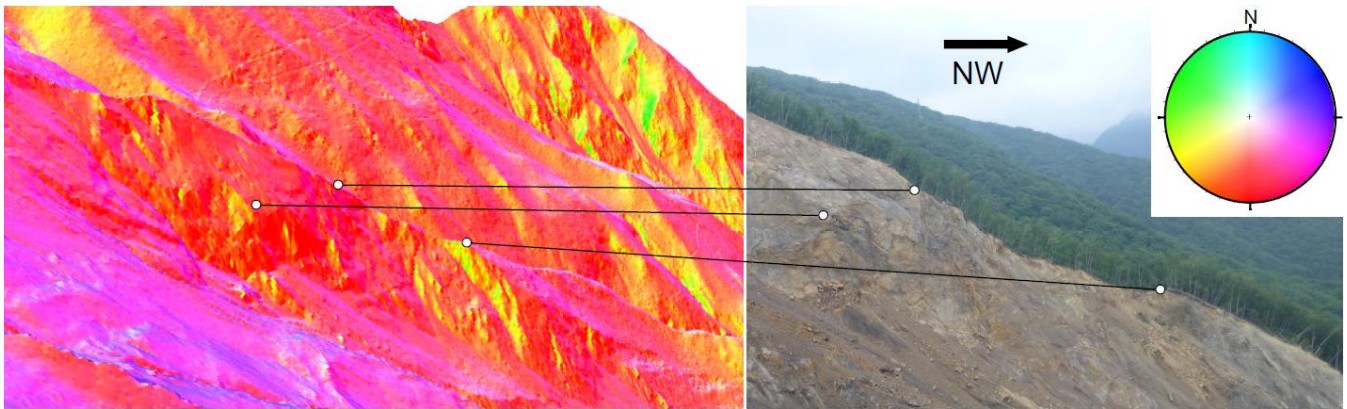

**Figure 18: Left: Correlation of the orientations from ArcScene-COLTOP scenery (the colours of the normal to the topography are given in the lower hemisphere) and on the right a picture of the southwestern upper scar of Aakatani landslide.**

### 4.4    Akatani-east landslide

The Akatani-east landslide presents clear crack development at the crest of the mountain; these cracks are visible on the pre-DEM hillshade. The post-DEM includes the landslide deposit that was eroded by the river. The volume deduced by Chigira et al. (2013) is $2.1 \times 10^6$ m³.

#### 4.4.1    Step 1: Two scenarios based on the pre-topography

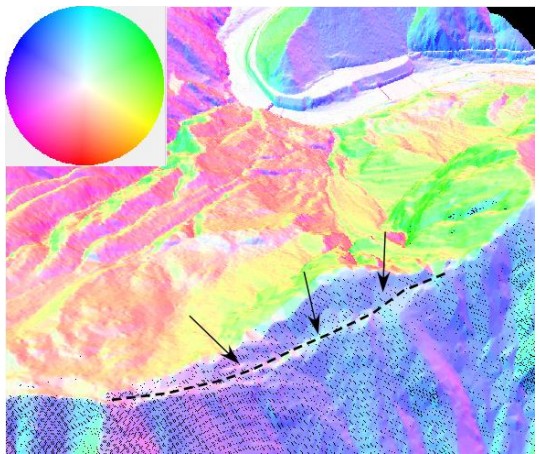

**Figure 19: Akatani-east landslide COLTOP-3D view of the counterscarp using the 1 m pre-DEM, indicating the upper expression of the failure surface, which extended from the sliding surface along a thrust (black dashed line and black arrows). The eastern margin (right in the figure) was bounded by a northwest-southeast trending and southwest-dipping joint, which appeared after the landslide. The resolution of the DEM is too low to provide an accurate slope.**

The first scenario is based on the pre-morphology deduced from the hillshade. The pre-event contour is based on the clear counterscarp at the top of the potential landslide, and the lateral limits are drawn based on the geomorphic features (Figure 19). The initial $C$ value is estimated using a longitudinal cross-section drawn by hand (Figure 20 and 21). The ratio $e$ for $z_{max}$

= 38 m and *L* = 710 m provides *C* = 0.0075. Taking into account that *e* used in Figure 20 is very small, it can be assumed that the second derivative is larger. In the second scenario, *C* = 0.015 is used. The results are compatible with the post-event topography (Figures 20 and 21). The cross-section is mostly in agreement with the post-DEM cross-section, but it shows an incorrect estimation at the top, where the thickness of the landslide is underestimated. The volumes obtained using the pre-DEM and the SLBL range from 1.59 and $2.06 \times 10^6$ m³.

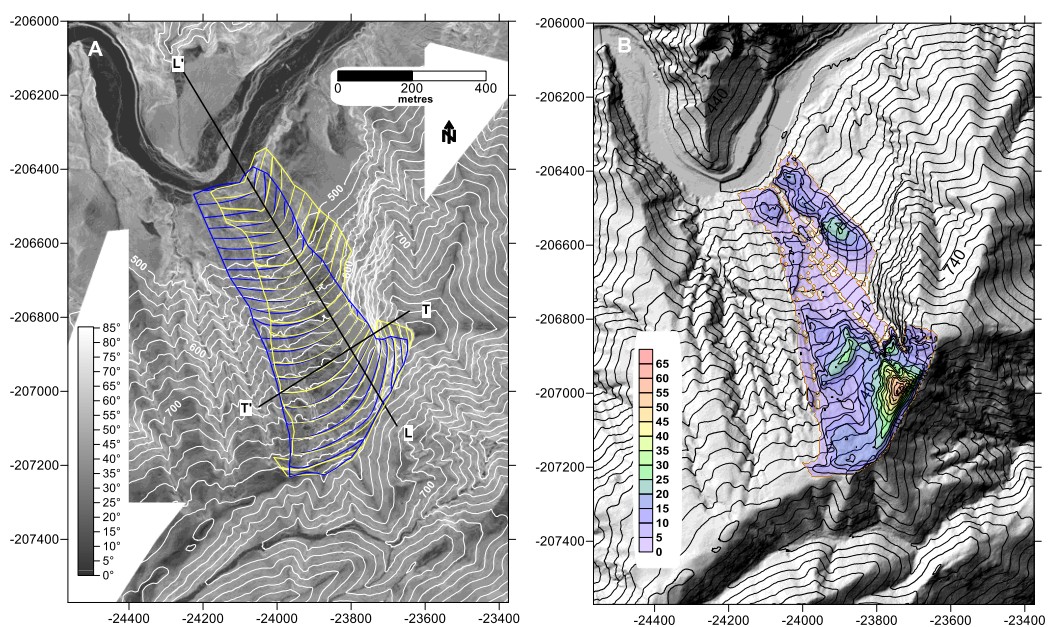

**Figure 20: A. Slope angles of the area around the Akatani-east landslide after the event (white contours). The blue contours correspond to scenario 1, and the yellow contours correspond to the mixed scenario. B: Hillshade of the pre-DEM with contours in black, displaying the thickness of the material in metres above the mixed scenario result. The cross-section is indicated.**

### 4.4.2    Step 2: Reconstruction of the failure surface using both the pre- and post-DEMs

The contour is defined using both the pre- and post-DEMs, which show that the western part of the slope was not as affected as the first interpretation would suggest. Using the same *C* values, the failure surface captures the post-DEM contours more accurately. Nevertheless, the disparity at the top of the scar remains (Figures 20 and 21). The volumes range from 1.75 to 2.15 $\times 10^6$ m³ for *C* = 0.0075 to *C* = 0.015, respectively.

### 4.4.3    Steps 3 and 4: Deposit reconstruction and volume estimations

Using identical *C* values for the shape of the failure surface (as a relevant way to estimate the surface of deposit, based on our experience), the missing part of the deposit was reconstructed using *C* = 0.015 with an inverse SLBL within a region defined by a polygon delineating the missing deposit based on the observations of the post-DEM hillshade (Figures 21 and 22). The difference between the reconstructed deposit DEM and the SLBL included in the pre-DEM for the second landslide contour provides volumes of 2.43 and $2.64 \times 10^6$ m³ for *C* = 0.0075 and *C* = 0.015, respectively. The thickness map of the second

SLBL clearly shows an underestimation (deficit) of approximately 0.5 Mm$^3$ at the top of the landslide (Figures 21 and 22). However, it seems that at the bottom of the slope, the SLBL is too deep. Therefore, both positive and negative volumes are probably balanced.

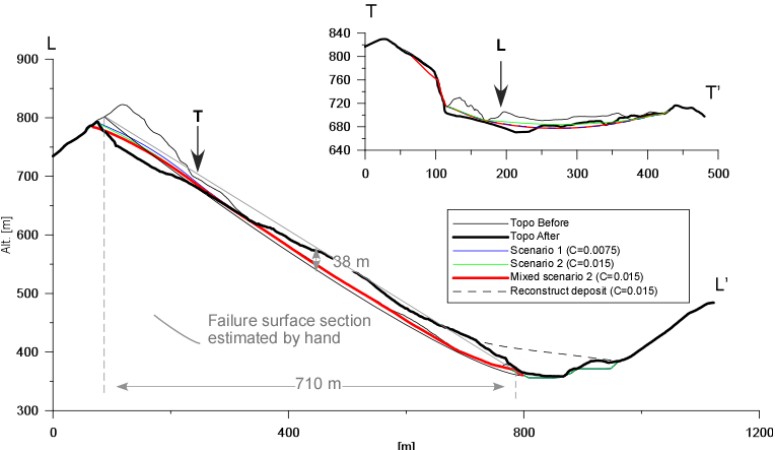

5    **Figure 21: Cross-section of the Akatani-east landslide, displaying different solutions of the SLBL and the deposit reconstruction. The cross-section made by hand is illustrated in grey.**

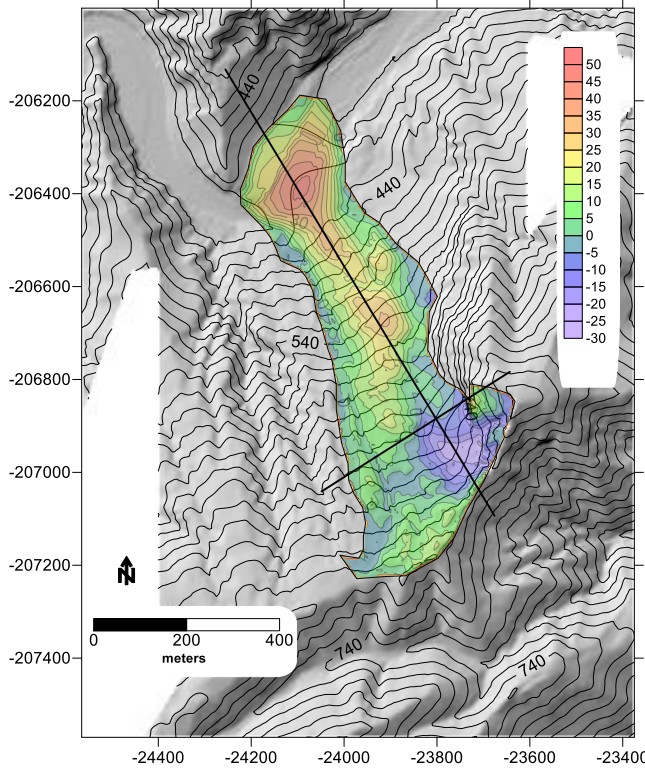

**Figure 22: Map of the thickness (in metres) of the reconstructed deposit of the Akatani-east landslide based on the inverse SLBL (C = 0.015), which was subtracted from the mixed scenario result. The background is the post-DEM hillshade and includes the**
10   **reconstruction of the lower part of the deposit.**

The estimations of the coefficient of expansion, not taking into account the SLBL deficit, are 38% for C = 0.015, which is unrealistic, and 22% for C = 0.0075. Adding the deficit to the volume of the landslide leads to approximately 3% for the expansion coefficient in both cases; this value is also unrealistic. An estimation of 22% is quite reasonable, but shows the large uncertainty in estimations of both the failure surface and the missing deposit, despite the fact that the volume estimation for *C* = 0.015 is close to that of Chigira et al. (2013).

### 4.4.4 Brief remarks

Steps 5 and 6 were not performed for this landslide because the pre-topography was already affected by slope movement and the valley floor is filled by river sediments.

The volume missed by the SLBL at the top of the landslide is certainly caused by the strong control of the surface by major structures, such as a regional thrust (as demonstrated by Arai and Chigira, 2018), faults and persistent joints sets. Multiple sets of wedges were formed by the possible structures that created the failure surface below the counterscarp, following the regional fault (Figure 19) and joints located in the eastern part of the upper landslide. The orientation of these structures is probably more to the west than the direction of the whole landslide. Therefore, major structures and/or persistent discontinuities that can invalidate the SLBL. However, the result of the SLBL gives a good approximation of the volume and the post-DEM at the top of the scar, except close the eastern crown where structures strongly control the failure surface.

## 4.5 Nagatono landslide
### 4.5.1 Step 1: Three scenarios based on the pre-topography

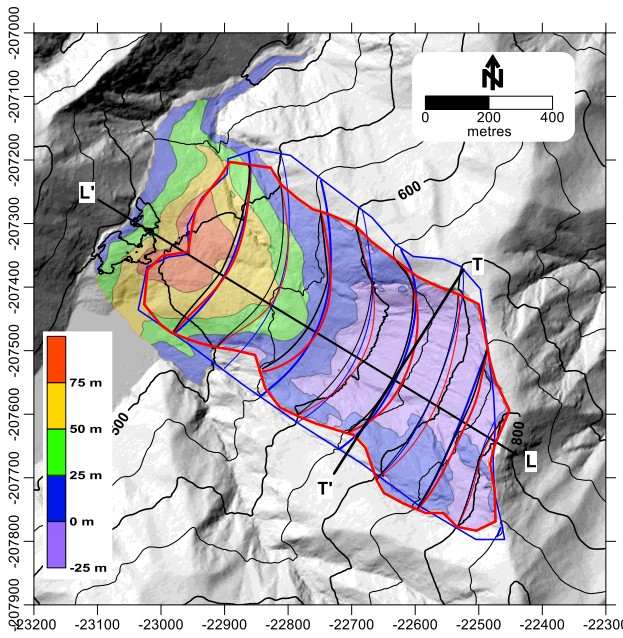

**Figure 23: Hillshade from the 1 m post-DEM of the Nagatono landslide with contour lines displaying the various results. In red, mixed scenario 2 (the preferred solution); in black, mixed scenario 1; and in blue, scenario 1. The coloured area refers to the thickness deduced from the subtraction of the post-DEM from mixed scenario 2.**

Based on the hillshade, we defined the limit of the potential landslide on the hillshade of the pre-DEM. In that case, the features and limits were quite obvious. The *a* value for scenario 1 was defined using a longitudinal cross-section on which a possible failure surface was drawn (Figures 23 and 24). *e* was evaluated at 0.11, which provides $C = 0.019$. The result is a volume of $3.94 \times 10^6$ m³. Chigira et al. (2013) found a volume of $4.1 \times 10^6$ m³; by modifying *C* to 0.02, we obtained the same result (Scenario 2: Table 1).

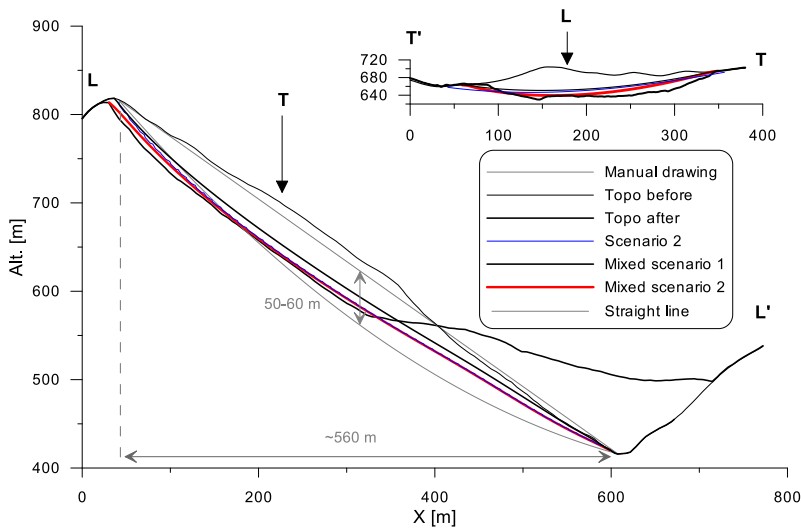

**Figure 24: Cross-section of the Nagatono landslide. The manual construction of the failure surface is given in grey.**

### 4.5.2     Step 2: Reconstruction of the failure surface using both the pre- and post-DEMs

The mixed scenario, based on a contouring of the changes before and after the landslide and assuming the failure surface outcrops at the base of the slope, led to only $3.19 \times 10^6$ m³ using $C = 0.02$. However, using $C = 0.03$, the result is $4.09 \times 10^6$ m³. The average thicknesses over the 162,287.50 m² surface are respectively 19.3 m and 24.8 m, which correspond to a difference of 5.5 m. The maps of the failure surfaces are not very different (Figure 23).

### 4.5.3     Step 3: Adding missing volumes

The contour of the deposit was rather simple, except near the lake. Based on a profile of the valley crossing the deposit, the volume below the water level was compensated by taking into account a few metres of lake area.

### 4.5.4     Step 4: Deposit volume calculations

Using the contour described above, and subtracting the mixed scenario DEM from the post-DEM, the total volume of sediments reaches 4.77 Mm³, which translates to an expansion coefficient of 19%.

### 4.5.5 Steps 5 and 6: Reconstruction of the topography before the event based on the post-event topography

An attempt to reconstruct the topography based on the post-DEM was not performed because the topography was subjected to other landslides after the main event and is deeply incised by a river and the associated civil works. In addition, it was too challenging to reconstruct the valley based on the post-DEM because several creeks disturbed the topography of the main valley, making it difficult to draw the correct limits.

### 4.5.6 Remarks

The resultant maps show that the limits of the Nagatono landslide deduced from the pre- and post-DEMs are very close to that of the actual event and that the limits fit the post-DEM in the upper area of the scar quite well (Figure 23). The comparison of the results with those of the pre-DEM also indicates a successful approach. The main problem may come from the fact that the upper part of the reconstructed sliding surface is not curved enough, while the lower part is possibly too curved. This is probably mainly caused by the local structures, such as faults.

## 5 Discussion

From the results obtained, the cross-sections clearly show that the SLBL can fit the failure surfaces well in areas without deposits. Often, the best SLBL results have parabolic shapes, as can be seen in Figures 6, 10, 14, 21 and 24; this result is rather intuitive when taking into account how simple the calculation is. In addition, a preliminary study indicates that cross-sections of failure surfaces observed in nature are well fitted by parabolas. In addition, in our experience, and as shown for the Kitamata mixed scenario, the use of a 1 or 5 m DEM does not change the results of the size of landslides considered in this study. This represents the first argument in favour of the simple SLBL method. Even if major and local structures can control the failure surfaces, as Chigira et al. (2013) and Arai and Chigira (2018) have shown, it appears that the final failure may be thrusted and that several faults and joint sets, which create a failure path and mimic parabolas, are present in the best results of the SLBL. Therefore, even if structures exist in the slope, on average, the failure follows a quadratic surface, as demonstrated by the many scar profiles that can be accurately fit using a parabola. When using the SLBL method, the main issue it to find the best value of $C$, which can be determined by using the ratio $e$ (ratio of the maximum thickness and length of the landslide), which provides rather good results.

The delineation of the landslide, especially before the landslide release, is a rather challenging problem; however, in the case of the Kii Mountain area, the limits are rather clear because the slope deformations are visible. However, real limits follow features that are not necessary identifiable at the surface before the landslide release and may also follow new segments as the intact rock breaks. Except for the Shimizu landslide, the limits of the different scenarios were close to those observed in nature, considering that the authors had no prior knowledge of where the real limits were located for the first step. Clearly, before a release, several scenarios have to be calculated to try to capture the uncertainty. When adding the knowledge of the post-DEM, the results are improved and usually closely follow the failure surface at the top of the landslide, especially where the scars are free from deposits.

The volume estimations before the release (Table 1; Figure 25) show that the results for scenario 1 are in agreement with the calculations of Chigira et al. (2013), except for the results of the Akatani landslide, where the assumed extent of the landslide was larger than the real extent. Nevertheless, the second attempts, considering the volumes, gave excellent results even for the Akatani landslide, for which a new contour was created. For the Shimizu landslide, no second scenario was calculated because the first profile obtained was reliable, yet the contour was including a too large area at the toe, creating an excess volume of 40%.

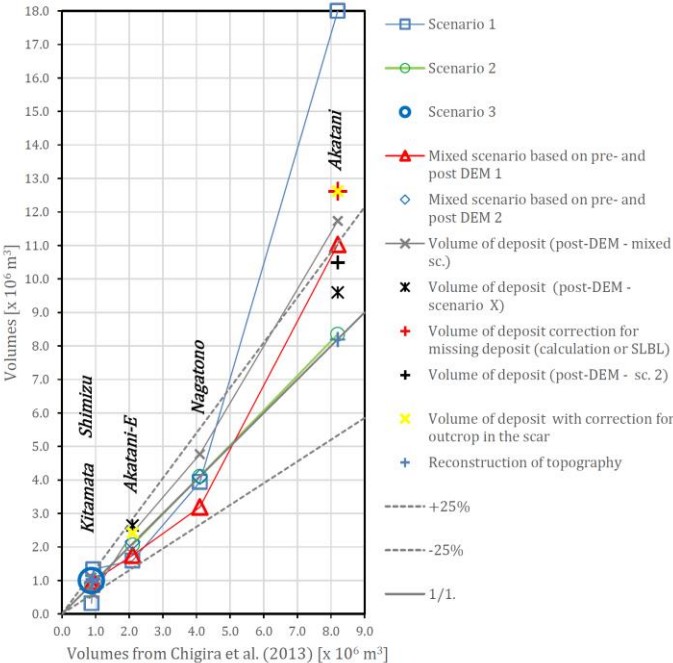

**Figure 25: All attempts to calculate the volumes of the five landslides in relation with the volume defined by Chigira et al. (2013).**

Based on the mixed scenario, the first version of the volume calculations are coherent with the estimation of Chigira et al. (2013), while the second attempt obtained better results for the Akatani-E and Nagatono landslides, thus demonstrating that the adjustments are rather simple to implement. The results for the other landslides were not recalculated because the cross-sections obtained with the first mixed scenarios were acceptable.

For the Kitamata, Schimizu and Akatani landslides, scenario 1 led to large volume differences because the estimation of their contours based on the pre-DEM interpretations of the geomorphic features were significantly different from the one of the real landslides. Nevertheless, these solutions can be considered as potential scenarios. Except for the three previous exceptions, all the volume calculations range between ±35% (Figure 25) of the estimation of Chigira et al. (2013).

Deposit volumes are not always easy to compute because of missing data on factors such as deposits hidden by a lake or erosion due to rivers. The reconstruction process is not simple, making reliable expansion coefficient results difficult to obtain. In the literature, the coefficient of expansion is often assumed to be 33% (Cruden and Varnes, 1996; Nicoletti and Sorriso-

Valvo, 1991), and recent landfill landslides had a coefficient of expansion of 8% (Yin et al., 2016). In our case, the most realistic expansion coefficient results range between 6% and 25%, the great variability for the same landslide depends on the scenarios considered (Table 1). This shows the great challenge of obtaining reliable information about such values. However, these results also provoke the question of whether expansion coefficients are linked to the slope deformation. It seems that 10-

25% of the expansion can be attributed to the increase in volume due to the release of the landslide and that the amount of expansion caused in situ by the slope deformation is in the range of approximately 0-8% to 15-18% assuming that the maximum total expansion coefficient (in situ + landslide release) cannot exceed 30 to 33%, but this needs further investigations.

If we look at the deduced values from a $C$ value equivalent to the constant of the squared term, three different behaviours that are not dependent on the scenarios (all are grouped for one landslide) can be identified (Figure 26). The small landslides of

Kitamata and Shimizu show a high second derivative ($2a$) compared to those of the larger landslides such as the Nagatono and Akatani-east landslides. The second derivative of the Akatani landslide seems to sit between all other solutions. The $e$ values range from 0.05 to 0.25 for all solutions, except for the Akatani solution, which had $e$ values that ranged up to 0.035. The difference of the Akatani landslide can be explained by how SLBL operates in 2.5D; it also takes into account the transversal profile. As a consequence, since the landslide is controlled by lateral faults and a thrust, the transversal profile has a higher $e$

value, but the $a$ value remains low because of the size of the landslide. The low $a$ value for the Nagotono and Akatani-east landslides is probably due to the strong control of thrusts or faults acting as sliding surfaces. Looking at the curvature values along the longitudinal profile of the most accurate mixed scenarios at the centre of the landslide, it seems that the Shimizu landslide is unique. The Shimizu landslide possesses a high curvature because there is strong structural control at the bottom of the slope. The Akatani, Nagatono and Kitamata landslides have very similar curvatures. For the Akatani-east landslide, this

comparison is irrelevant because the SLBL does not fit the top of the landslide well.

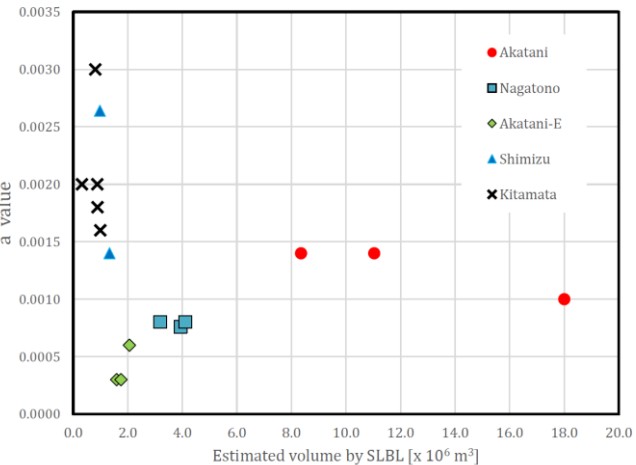

**Figure 26: Values of $a$, the equivalent parabola term from equation 6, in relation to the volumes estimated for different scenarios of each landslide.**

From the above discussion, it is clear that SLBL will not give "the solution" for a future landslide, but it can provide an efficient tool for quickly proposing several scenarios of volumes that can be released by an identifiable slope deformation or an expected landslide. This must be coupled with a landslide specialist knowledge. It is also clear that this approach must utilize the results of the analysis of the available data, which lead to a conceptual model that will in turn constrain the SLBL. As an example,

this method has been used by Pedrazzini et al. (2012) to estimate the potential deposit volume that can produce rock avalanches from the South Peak of Turtle Mountain (Frank Slide). These volumes and geometry were used to perform runout simulations, underlining the importance of volume estimation because of the well-known dependence of travel distance on volume. Concerning the palaeotopography reconstruction, the results show that this method can efficiently fill missing volume due to erosion but is strongly dependent on the knowledge of the limits of the deposits. It is clear that some erosion tools must be

added to improve the prediction of the topography before a landslide. To reduce the artefacts produced by the SLBL in order to reconstruct the ancient topography of buried fluvial valleys, one solution would be to proceed by modifying the original post-DEM by drawing a hypothetical river in the central part of the valley and replacing the altitude of pixels of the post-DEM along the river path by a linear interpolation of the altitude joining the points that are close to the deposit but not affected by it. Thus, the SLBL will deepen the topography to that level, assuming that the river pixels have fixed altitudes.

Such an approach is still fastidious and long; clearly, it will be advantageous to develop a software that integrates all these capabilities to easily perform trial and error analyses, test hypotheses, produce cross-sections, and add other constraints such as faults, fixed points, rivers, and boreholes. In addition, in this study, we did include the simple additional constraint of limiting the slope angle for the SLBL, opening several other possibilities regarding landslides constrained by a sliding plane or known geomechanical friction angle limit.

**6    Conclusions**
The data from the landslides triggered by Typhoon Talas on the Kii peninsula were used to test the SLBL techniques. The results show that volumes are estimated within an acceptable range, ±35%, for most of the realistic scenarios. However, this range is strongly dependent on identification of the geomorphic features delineating the future landslides. The SLBL method also allows the estimation and reconstruction of the deposit volumes, showing that this procedure is quite complex and that

the expansion coefficients are difficult to estimate.
The failure surfaces obtained using the SLBL method generally agree with the failure surface observed in the post-DEM within the area free of the deposit. The results are improved when (1) they are adjusted to obtain volume estimates similar to those deduced by Chigira et al. (2013) and (2) the contours of the landslides used come from an interpretation of both the pre- and post-DEMs (mixed scenario). To obtain the expansion coefficient of some of these landslides, the volumes of the deposits

missing due to river erosion were calculated using an inverse SLBL. The coefficients of expansion deduced from the reconstructions range from 5% and 25%. The reconstructions of the topography before the landslides, in the scar or below the deposits not only give reliable results but also show the impact of erosion due to rivers and local slope movements after the main slide.

A SLBL can be a valuable tool to quickly produce a 2.5D failure surface for a given landslide, knowing that the volume increases linearly with C value depending on the landslide limits, which needs further investigations. Several additional scenarios must be created, based on all the available knowledge of the landslide, indicating that quadratic surfaces are a good compromise for fitting failure surfaces, even if the landslide failure surfaces are mainly controlled by structures (thrust, faults,

joints, bedding, etc.). Therefore, when the failure surface is not controlled by a unique structure, the failure surface mimics a quadratic surface because the network created by structures tend to define quadratic shapes when combined. Consequently, the SLBL calculation seems to be a suitable solution to fit such surfaces.

Adding other constraints (slope, defined surfaces, etc.) to the calculation of the SLBL can greatly improve the results. To be efficient, these constraints must be integrated in a tool that permits quick execution of this method in order to produce and

manage multiple scenarios.

**Acknowledgements**

This project was funded and supported by the Disaster Prevention Research Institute, Kyoto University, by supporting the first author's visit at the Uji campus in the summer of 2015. The first author is profoundly grateful for this support.

The authors are grateful for the support of the Nara Prefectural Government and the Kinki Regional Development Bureau of

the Ministry of Land, Infrastructure, Transportation, and Tourism, who provided the DEMs. The English language of this manuscript was edited by American Journal Experts, and we thank them for their support. We thank Alexander Strom and an anonymous reviewer for their fruitful comments.

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
