# Peer review of "Testing a failure surface prediction and deposit reconstruction method for a landslide cluster that occurred during Typhoon Talas (Japan)"

_Earth Surface Dynamics, 2018_

## Referee Comment (RC1) · Anonymous Referee #1 · 26 Aug 2018

This paper presents the capability of the Sloping Local Base Level (SLBL) method to provide models of the 3D failure surfaces of landslides based on digital elevation models, as well as to reconstruct buried valley topographies and landslide deposit surfaces.

The proposed method has been tested on 5 deep-seated landslides that occurred during Typhoon Talas, which hit Japan in August 2011.

The topics covered in this paper are extremely interesting for those involved in landslide practice, since landslide volume estimation is very often a challenging task, and its fast and accurate estimation is fundamental for the definition of reliable risk scenarios, especially in emergency conditions.

[Figure]

In the manuscript different procedures and steps are proposed to assess the volumes, the failure surface and the palaeotopography. Although these are generally well explained, they seem not to be easy to apply, as the elaboration output are very sensitive to some parameters that need to be assumed by the users. For example, the Authors adjusted several times the tolerance parameter 'C', in order to obtain the desired results.

The application to the 5 case studies is very interesting, and it allows comparing the different elaborations depending on the characteristics of the investigated phenomena. The discussion of the results is clear and well organized.

For these reasons, in my opinion, the paper is worth to be published.

Minor remarks about images:

Sometimes the font size within the figures is too small. Please, enlarge it, to make the images more legible

Figure 8 is not clear and legend is missing

---

## Referee Comment (RC2) · A. Strom (Referee) · 15 Sep 2018

Dear Authors,

Thank you for interesting paper. I recommended to accept it after minor revisions due to following reasons. Some statements need additional clarification. 1) What do you mean as 2.5D surface (in the Abstract and in Conclusions). Short explanation how it differs from 3D will be useful for the reader. 2) in section 1.2 it will be good if you will add basic linear dimensions of the described landslides (length, width, slope height). 3) It is important to notice that L is the horizontal projection of the sourse zone length, not total runout that is described in most of paper as the horizontal projection

of the distance between the headscarp crown and the deposits tip. While L can be defined in a univocal manner, W (width) definition needs clarification - is it a maximal or mean value. 4) Step's order (Figure 3, and in the text). For me it remains uncleare why do you start from the pre-DEM. It looks more logical (at least for landslides that have occurred already) to start from the post-DEM where we know exactly what are the landslide dimensions, at least in the upper part of the headscarp. May be some additional explanation is necessary. 5) Figure 7. Add, please, what are A, B, and C. 6) Figure 8, 16, 17. Legend will be usefu (as on Fig. 18). 7) Page 27, lines 29-30. Statement that that the amount of expansion caused in situ by the slope deformation is in the range of 8% to 23% require some comments. It seems to be too large (before real release of landslide).

---

## Author Comment (AC1) · 9 Nov 2018

Dear referee #1 and Dear editor,

Thank you for the nice comments, and recommendation for publication.

The remark about C-value is true. We are aware that C is difficult to choose. For that reason we will add some comments about volume changes with C-value changes all along the text. .

We will improve the figure 8 by adding the colour scheme and the legend and provide another 3D point of view.

[Figure]

We will enlarge the small fonts in the figures and in some cases we will make the figures larger.

---

## Author Comment (AC2) · 9 Nov 2018

Dear referee #2 and Dear editor,

1. The 2.5D corresponds to a surface which possess for each x-y coordinates, one and only one z value, in other words, no true vertical topography or overhang can be represented perfectly.

2. We will add in section 1.2 the width, length and slope height of the landslides.

3. We will change L to Lrh and referring to horizontal projection of failure surface length Lr defined by Cruden and Varnes (1996). The width correspond to the width

along defined cross-sections, in order to evaluate the curvature.

4. Here we always starts by the pre-DEM, because the first author has tried to define the landslide contours and volumes without knowing the contour given by the post-DEM. We will clarify that point along the text. This is performed to illustrate the potentiality of the method to define different scenarios for the failure surface and the volume, which may be involved in a future catastrophic failure.

5. The full legend will be added to figure 7, it is missing sorry.

6. We will add the colour scheme on figures 8, 16, and 17 and for the figure 8 the 3D view will be improved.

7. We will give more explanation about the 8-23% for "in situ expansion". You are right the argument are missing. It comes from the fact that we assume an average expansion for the deposit from which we removed the expansion caused by the catastrophic release of the landslide ((volume of the deposit – volume "in situ" of the instability)/volume of the deposit). It is clear that the maximum value is large, but looking at some catastrophic slope deformations, this may be possible.

---

## Author Response (AR1)

- **The major changes are highlighted in yellow in the manuscript**
- **Remarks of the reviewers in bold and they are highlighted in yellow in the below answers.**

**Reviewer 1 answer**

Dear referee #1 and Dear editor,

Thank you for the nice comments, and recommendation for publication.

**This paper presents the capability of the Sloping Local Base Level (SLBL) method to provide models of the 3D failure surfaces of landslides based on digital elevation models, as well as to reconstruct buried valley topographies and landslide deposit surfaces. The proposed method has been tested on 5 deep-seated landslides that occurred during Typhoon Talas, which hit Japan in August 2011. The topics covered in this paper are extremely interesting for those involved in landslide practice, since landslide volume estimation is very often a challenging task, and its fast and accurate estimation is fundamental for the definition of reliable risk scenarios, especially in emergency conditions. In the manuscript different procedures and steps are proposed to assess the volumes, the failure surface and the palaeotopography. Although these are generally well explained, they seem not to be easy to apply, as the elaboration output are very sensitive to some parameters that need to be assumed by the users.**

**For example, the Authors adjusted several times the tolerance parameter 'C', in order to obtain the desired results.**

The remark about C-value is true. We are aware that C is difficult to choose. For that reason we added an example of how C is related to volume.

**Minor remarks about images: Sometimes the font size within the figures is too small.**

We will enlarge the small fonts in the figures and in some cases we will make the figures larger.

**Please, enlarge it, to make the images more legible Figure 8 is not clear and legend is missing**

We will improve the figure 8 by adding the colour scheme and the legend and provide another 3D point of view.

**Reviewer 2 answer**

Dear referee #2 and Dear editor,

Thank you for the nice comments, and recommendation for publication.

**1) What do you mean as 2.5D surface (in the Abstract and in Conclusions). Short explanation how it differs from 3D will be useful for the reader.**

1. The 2.5D corresponds to a surface which possess for each x-y coordinates, one and only one z value, in other words, no true vertical topography or overhang can be represented perfectly. It is added in the abstract.

**2) in section 1.2 it will be good if you will add basic linear dimensions of the described landslides (length, width, slope height).**

2. We will add in section 1.2 the width, length and slope height of the landslides.

**3) It is important to notice that L is the horizontal projection of the sourse zone length, not total runout that is described in most of paper as the horizontal projection of the distance between the headscarp crown and the deposits tip. While L can be defined in a univocal manner, W (width) definition needs clarification - is it a maximal, or mean value.**

3. We will change L to Lrh and referring to horizontal projection of failure surface length Lr defined by Cruden and Varnes (1996). The width correspond to the width along defined cross-sections, in order to evaluate the curvature.

**4) Step's order (Figure 3, and in the text). For me it remains uncleare why do you start from the pre-DEM. It looks more logical (at least for landslides that have occurred already) to start from the post-DEM where we know exactly what are the landslide dimensions, at least in the upper part of the headscarp. May be some additional explanation is necessary.**

4. Here we always starts by the pre-DEM, because the first author has tried to define the landslide contours and volumes without knowing the contour given by the post-DEM. We will clarify that point along the text. This is performed to illustrate the potentiality of the method to define different scenarios for the failure surface and the volume, which may be involved in a future catastrophic failure. The step two takes advantage of both DEM information.

**5) Figure 7. Add, please, what are A, B, and C.**

5. The full legend is added to figure 7, it was missing sorry.

**6) Figure 8, 16, 17. Legend will be usefull (as on Fig. 18).**

6. We will add the colour scheme on figures 8, 16, and 17 and for the figure 8 the 3D view will be improved.

**Page 27, lines 29-30. Statement that that the amount of expansion caused in situ by the slope deformation is in the range of 8% to 23% require some comments. It seems to be too large (before real release of landslide).**

7. We give more explanation about the 8-23% for "in situ expansion". You are right the argument are missing. It comes from the fact that we assume an average expansion for the deposit from which we removed the expansion caused by the catastrophic release of the landslide ((volume of the deposit – volume "in situ" of the instability)/volume of the deposit). It is clear that the maximum value is larger, but looking at some catastrophic slope deformations, this may be possible. But the values given were a mistake we modified to clarify: "It seems that 10-25% of the expansion can be attributed to the increase in volume due to the release of the landslide and that the amount of expansion caused in situ by the slope deformation is in the range of approximately 0-8% to 15-18% assuming that the maximum of total expansion coefficient (in situ + landslide release) cannot exceed 30 to 33%, but this needs further investigations."

---

## Author Response (AR2)

**Answers to the editor**

Dear editor,

Thank you for your recommendation and suggestions.

The changes that have been made are highlighted in yellow in the manuscript and in correction mode.

We did the following changes:

- As you requested we made some English language corrections, especially in the part that have been added by the review, because they were not validated by the American Journal expert. You can find several of them in correction mode and yellow in the new document.
- Because two first formulas were not numbered we renumbered the equations.
- The equations 6 and 11 were corrected for mistakes (old numbers 4 and 9).
- Page 7 line 19 we added a precision about the k value: elongation of landslides. … k = 1 is implies that it uses the average diameter instead of $L_{rh}$, and by assuming an elliptic horizontal landslide surface $k = \sqrt{\pi w/(4 L_{rh})}$.
- In the last paragraph of p 7 we change the fonts of the variables that were not in italics.

We hope that is all you requested.

Sincerely yours

Michel Jaboyedoff

---

## Author Response (AR3)

**Answers to the editor**

Dear editor,

Thank you for your recommendations and suggestions.

==The changes you requested for the final version have been made and they are highlighted in green in the corrected version of the manuscript.==

We did a few additional changes:

- Adding indication about orientation on figure 18 and 19.
- Some missing numbers have been added to Table 1.

We provide figure in pdf or JPG and the able both in pdf and excel. ==I was not able to insert the table 1 in text format in the manuscript, it is an image now. But I provide 3 versions in word (A3) , Excel and PDF. I hope it works.==

We hope that is all you requested.

Sincerely yours

Michel Jaboyedoff